# Emergent generalized symmetries in ordered phases

Salvatore D. Pace

*Department of Physics, Massachusetts Institute of Technology, Cambridge, MA 02139, USA*
(Dated: September 29, 2023)

We explore the rich landscape of higher-form and non-invertible symmetries that emerge at low energies in generic ordered phases. Using that their charge is carried by homotopy defects (i.e., domain walls, vortices, hedgehogs, etc.), in the absence of domain walls we find that their symmetry defects in $D$-dimensional spacetime are described by $(D-1)$-representations of a $(D-1)$-group that depends only on the spontaneous symmetry-breaking (SSB) pattern of the ordered phase. These emergent symmetries are not spontaneously broken in the ordered phase. We show that spontaneously breaking them induces a phase transition into a nontrivial disordered phase that can have symmetry-enriched (non-)abelian topological orders, photons, and even more emergent symmetries. This SSB transition is between two distinct SSB phases—an ordinary and a generalized one—making it a possible generalized deconfined quantum critical point. We also investigate the 't Hooft anomalies of these emergent symmetries and conjecture that there is always a mixed anomaly between them and the microscopic symmetry spontaneously broken in the ordered phase. One way this anomaly can manifest is through the fractionalization of the microscopic symmetry's quantum numbers. Our results demonstrate that even the most exotic generalized symmetries emerge in ordinary phases and provide a valuable framework for characterizing them and their transitions.

## I. INTRODUCTION

Recent generalizations of symmetry have refortified the power of symmetry to non-perturbatively characterize the dynamics and phases of many-body systems [1–4]. The crux of these generalizations stems from the modern perspective that a topological defect[1] always generates a symmetry [7]. In Lorentzian spacetime, a topological defect within a fixed time slice is the symmetry operator that commutes with the Hamiltonian and acts on the Hilbert space. When a topological defect extends in the time direction, it is a symmetry defect that modifies the Hilbert space.

From the point of view of topological defects, in $D$ dimensional spacetime, ordinary symmetries are generated by invertible codimension 1 topological defects $\{T_{D-1}^{(g)}\}$. The symmetry group $G$ describes their parallel fusion

$$T_{D-1}^{(g)} \times T_{D-1}^{(h)} = T_{D-1}^{(gh)}, \qquad g, h \in G. \qquad (1)$$

They are invertible topological defects because for each $T_{D-1}^{(g)}$ there exists a topological defect labeled by $h = g^{-1}$ whose fusion with $T_{D-1}^{(g)}$ yields the trivial defect $T_{D-1}^{(1)}$.

A fruitful avenue for generalizing ordinary symmetries is to modify the properties of $\{T_{D-1}^{(g)}\}$. For instance, there are topological defects that are not $(D-1)$-dimensional. So, one can generalize symmetries by having codimension $p+1$ topological defects ($0 \leq p \leq D-1$) also generate symmetries, which are called $p$-form symmetries [7–12]. When $p > 0$, $p$-form symmetries are called higher-form symmetries, and their symmetry charge is carried by extended objects of dimension $\geq p$. With this modification, symmetries are no longer generally described by groups

---

[1] Defects in spontaneous symmetry breaking (SSB) phases characterized by the topology of the order parameter—vacuum—manifold are sometimes called topological defects [5, 6]. However, these defects are generally not topological in the sense that they can be continuously deformed without modifying observables. In this paper, we will call these homotopy defects and reserve the term topological defect to describe defects that can be continuously deformed.

but instead by a kind of higher category called a higher-group [13–17].

Another interesting proposed generalization is to consider non-invertible topological defects as symmetries, which are called non-invertible symmetries [18–25]. A topological defect $T^{(a)}$ is non-invertible if there does *not* exist a topological defect $T^{(a^{-1})}$ for which $T^{(a)} \times T^{(a^{-1})} = T^{(1)}$. Evidently, non-invertible symmetries cannot be described by (higher-)groups, where each element is required to have an inverse, and are instead described by more general (higher-)categories [26–31] (see appendix B). For this reason, non-invertible symmetries are sometimes called (higher-)categorical symmetries.

Another generalization, which we will not consider in this paper, is to no longer require the topological defects to be fully topological. These partially topological defects are topological only within some subspace of space-time and generate symmetries called subsystem symmetries [32–37]. In a quantum theory, something that generates a symmetry must always be topological in the time direction to ensure that its symmetry operator commutes with the Hamiltonian. Therefore, subsystem symmetries can only occur in non-relativistic systems.

A symmetry can enjoy any combination of these and other generalized properties. With these ones, a generic symmetry is an $a_1$-$a_2$-$a_3$ symmetry, where the adjectives

$$a_1 \in \{\text{invertible, non-invertible}\},$$
$$a_2 \in \{0\text{-form}, 1\text{-form}, \cdots, (D-2)\text{-form}, (D-1)\text{-form}\},$$
$$a_3 \in \{\text{fully topological, subsystem}\}.$$

Furthermore, a system's total symmetry can include any mixture of ordinary and generalized symmetries. Including the higher-form and non-invertibility properties generalizes the fusion rule Eq. (1) to

$$T_{D-p-1}^{(a_p)} \times T_{D-p-1}^{(b_p)} = \sum_{c_p} N_{a_p b_p}^{c_p} \, T_{D-p-1}^{(c_p)}. \qquad (2)$$

If the sum on the right-hand side always includes only one term, the topological defects generate an invertible $p$-form symmetry; otherwise, they generate a non-invertible $p$-form symmetry. The sum is at the level of correlation functions. However, another way to interpret it is by inserting the topological defect $T_{D-p-1}^{(a_p)} \times T_{D-p-1}^{(b_p)}$ in the time direction to modify the Hilbert space $\mathscr{H}$ to the defect Hilbert space $\mathscr{H}_{T^{(a_p)} \times T^{(b_p)}}$. The fusion rule then implies

$$\mathscr{H}_{T^{(a_p)} \times T^{(b_p)}} = \bigoplus_{c_p} N_{a_p b_p}^{c_p} \, \mathscr{H}_{T^{(c_p)}}. \qquad (3)$$

This also makes it clear that $N_{a_p b_p}^{c_p}$ must be non-negative integers.

At the level of kinematics, generalized symmetries provide an economical and unifying organization principle for many-body systems. However, their real power lies in their ability to characterize and constrain a system's dynamics. For instance, generalized symmetries can characterize its phases by spontaneously breaking, giving rise to topological order, fractons, emergent photons, and other exotic phenomena [38–48], and by forming symmetry protected topological (SPT) phases [21, 48–63]. Furthermore, they can have 't Hooft anomalies [52, 64–71] and provide other constraints on renormalization group flows [20, 72–75] and out-of-equilibrium phenomena [76–78]. These powerful consequences also justify why topological defects should be interpreted as generalized symmetries: they can do what ordinary symmetries can do.[2]

At the microscopic scale, many-body systems will typically not have generalized symmetries. Indeed, microscopic models of condensed matter describe nuclei and electrons and will include generic two-body terms that explicitly break any generalized symmetries. From a high-energy physics point of view where the microscopic scale is governed by quantum gravity, it is believed there are no symmetries at all [79], including higher-form [80] and non-invertible symmetries [81, 82]. However, generalized symmetries can emerge at low energies/long distances and at critical points, which is why they are useful despite not generically being microscopic symmetries.

While ordinary symmetries that emerge at low energies are typically only approximate symmetries, emergent higher-form symmetries are exact at low energies [83–89]. This makes them just as powerful at low energies as exact microscopic symmetries. They can spontaneously break, have 't Hooft anomalies, characterize many-body phases and transitions, and provide general non-perturbative low-energy constraints [88].

It is, therefore, important to identify generic settings where generalized symmetries emerge and to understand their physical consequences. In this paper, we show that ordinary ordered phases, arising from spontaneously breaking ordinary symmetries, have emergent generalized symmetries at low energies. By low energy, we will always mean energy scales below any gapped excitations (i.e., the deep IR). Examples of such symmetries have been investigated previously in particular ordered phases [7, 39, 67, 90–94]. Here we explore the rich landscape of generalized symmetries in generic ordered phases in arbitrary dimensions, finding they can have any combination of non-invertible and higher-form properties and uncovering some of their general physical consequences. While we focus our attention on generic ordinary ordered phases, our results can be generalized to any phase with symmetries—generalized or ordinary— spontaneously broken. Furthermore, we will not specialize to any particular class of models as our arguments apply to any physical system with the SSB pattern. However, we note that our results generally apply to nonlinear sigma

―――――

[2] *If it looks like a duck, swims like a duck, and quacks like a duck, then it probably is a duck.*

models whose target space is the order parameter manifold of the SSB pattern.

The general feature of ordered phases we use to reveal these emergent generalized symmetries is the existence of topologically protected defects, which we will call homotopy defects (i.e., domain walls, vortices, and hedgehogs). Homotopy defects are generally not topological defects as defined by footnote 1. They play an important role in a wide range of fields in physics, from condensed matter [95] to cosmology and particle physics [5]. We discuss their classification in Sec. II, reviewing standard aspects in II A and presenting a new technique to represent them as magnetic defects (i.e., gauge fluxes) of a $(D-1)$-group $\mathbb{G}_\pi^{(D-1)}$ gauge theory in II B. At the level of the homotopy defects, the classifying space of $\mathbb{G}_\pi^{(D-1)}$ is the $(D-1)$th Postnikov stage of the order parameter manifold.

In physical systems, homotopy defects have dynamics and there can exist degrees of freedom on which they end. For example, in a $D = 3$ superfluid, the homotopy defects are vortex lines classified by the winding number and they can end on gapped particles. In Sec. III, we show that at energies below the gap of the degrees of freedom homotopy defects can end on, homotopy defects are detected by topological defects. Therefore, there is an emergent symmetry whose symmetry charges are carried by the homotopy defects and correspond to their classification. In the absence of domain walls, at energy scales below any gapped degrees of freedom, we show that the emergent symmetry's symmetry defects are described by the $(D-1)$-representations of $\mathbb{G}_\pi^{(D-1)}$. When there are finitely many classes of homotopy defects, this means that the fusion $(D-1)$-category describing this generalized symmetry is $\mathcal{S} = (D-1)\text{-Rep}(\mathbb{G}_\pi^{(D-1)})$. We verify our result by independently deriving it from the homotopy defect's classification using the Symmetry TFT (SymTFT). We discuss this symmetry category and SymTFT construction for particular examples in Secs. III A–III D.

In the ordered phase, these emergent generalized symmetries are not spontaneously broken. In Sec. IV, we argue that spontaneously breaking them induces a phase transition to a nontrivial disordered phase with (non-)abelian topological order, emergent photons, and even more emergent generalized symmetries. The critical point is controlled by the microscopic symmetries and the emergent symmetries common to both the ordered and nontrivial disordered phases. It is a direct transition between two SSB patterns—an ordinary one and a generalized one—and, therefore, is possibly a type of generalized deconfined quantum critical point. We demonstrate the existence of such phase transitions and additional emergent symmetries in simple examples using Euclidean lattice models in Secs. IV A and IV B.

These emergent generalized symmetries can also have 't Hooft anomalies, which we discuss in Sec. V. The fact that the homotopy defects of the microscopic symmetry are charged under the emergent generalized symmetry strongly suggests there is a mixed anomaly between them. We conjecture that such a mixed 't Hooft anomaly always exists. We provide evidence for this conjecture through physical reasoning and by showing examples in V A.

## II. HOMOTOPY DEFECTS

Consider a system in $D$-dimensional Euclidean spacetime $M_D$ with an internal ordinary symmetry described by the group $G$ that is spontaneously broken to $H \subset G$. The different SSB patterns are labeled by the conjugacy classes of the subgroups $H$ and are characterized by a local order parameter $\mathcal{O}(x)$ that is an $H$ singlet. When space is path-connected, the ground states are labeled by elements of the order parameter manifold[3]

$$\mathcal{M} = \{g\mathcal{O} \mid \forall g \in G/H\} \simeq G/H. \tag{4}$$

### A. The order parameter presentation

The details of $\mathcal{O}$ depend on microscopic details. For our purposes, it is convenient to consider one that takes values in $\mathcal{M}$. When $\mathcal{M}$ is discrete, we can triangulate $M_D$ and consider $\mathcal{O}(x)$ as a lattice field instead of a continuum field. This makes $\mathcal{O}$ a map from $M_D$ to $\mathcal{M}$. Therefore, for any closed $k$-submanifold $\Sigma_k$ of $M_D$, each configuration of $\mathcal{O}(x)$ belongs to an equivalence class of

$$[\mathcal{O}|_{\Sigma_k}]_{\text{f}} \in [\Sigma_k, \mathcal{M}]_{\text{f}}, \tag{5}$$

the set of maps from $\Sigma_k$ to $\mathcal{M}$ up to free homotopy. Furthermore, we equip $\Sigma_k$ and $\mathcal{M}$ with base points $s$ and $m$, respectively, and consider based maps (i.e., $\mathcal{O}(s) = m$). Then, Eq. (5) is the set of based maps up to free homotopy.

The order parameter represents the system's configuration and is the constant map $\mathcal{O}_{\text{gs}}$ for a ground state—a map to a single point in $\mathcal{M}$. If $[\mathcal{O}|_{\Sigma_k}]_{\text{f}} \neq [\mathcal{O}_{\text{gs}}]_{\text{f}}$, $\mathcal{O}(x)$ represents a configuration that cannot be continuously deformed to a ground state in the minimal volume $(k+1)$-manifold $B_{k+1}$ with $\partial B_{k+1} = \Sigma_k$. Such a configuration is interpreted as having a defect, where $\mathcal{O}(x)$ is singular, intersecting $B_{k+1}$ that is responsible for this obstruction [95]. We will call this defect a homotopy defect (see footnote 1) and denote the operator that inserts one on the $n$-submanifold $C_n$ as $H(C_n)$.

Homotopy defects that can be detected using $\Sigma_k$ are classified by Eq. (5), and the equivalence class $[\mathcal{O}|_{\Sigma_k}]_{\text{f}}$ is often referred to as the charge of the homotopy defect. The most commonly studied ones are those detected by

———

[3] For a $p$-form symmetry $G^{(p)} \xrightarrow{\text{ssb}} H^{(p)}$, the order parameter manifold is $\mathcal{M} = B^p(G/H) \simeq B^pG/B^pH$, constructed by delooping $G/H$ $p$ times [14, 96].

$\Sigma_k \simeq S^k$ through linking and are codimension $k+1$. For this simple class of defects, it is convenient to consider instead the $k$th homotopy group

$$\pi_k(\mathcal{M}) \equiv [S^k, \mathcal{M}]_{\mathrm{b}}, \qquad (6)$$

which is the set of maps from $S^k$ to $\mathcal{M}$ up to based homotopy. While freely homotopic maps may not be based homotopic, there is an action of $\pi_1(\mathcal{M})$ on $\pi_k(\mathcal{M})$ that connects freely homotopic elements of $\pi_k(\mathcal{M})$ and provides the one to one correspondence[4]

$$\pi_k(\mathcal{M})/\pi_1(\mathcal{M}) \leftrightarrow [S^k, \mathcal{M}]_{\mathrm{f}}. \qquad (7)$$

For instance, the action of $\pi_1(\mathcal{M})$ on itself is by conjugation, so these codimension 2 homotopy defects are classified by the conjugacy classes $\mathrm{Cl}(\pi_1(\mathcal{M}))$ of $\pi_1(\mathcal{M})$. More generally, these actions are described by the group homomorphisms

$$\alpha_n : \pi_1(\mathcal{M}) \rightarrow \mathrm{Aut}(\pi_n(\mathcal{M})), \qquad (8)$$

where $\mathrm{Aut}(\pi_n(\mathcal{M}))$ is the group of automorphisms of $\pi_n(\mathcal{M})$.

When the action of $\pi_1(\mathcal{M})$ on $\pi_k(\mathcal{M})$ is trivial, $\pi_k(\mathcal{M}) \simeq [S^k, \mathcal{M}]_{\mathrm{f}}$ and these homotopy defects are characterized by $\pi_k(\mathcal{M})$. This includes their fusion

$$H^{(g)} \times H^{(h)} = H^{(gh)}, \qquad g, h \in \pi_k(\mathcal{M}), \qquad (9)$$

from which it is clear they are invertible defects (see Fig. 1).

Since homotopy defects detected by $S^k$ are codimension $k+1$, homotopy defects classified by $\pi_k(\mathcal{M})$ for $k > D-1$ are not present in $D$ dimensional spacetime. Therefore, only the $(D-1)$-homotopy type of $\mathcal{M}$ matters when classifying homotopy defects. When $\mathcal{M}$ is connected and admits a CW-decomposition, we can truncate it to $\mathcal{M}_n$ which satisfies

$$\pi_k(\mathcal{M}_n) = \begin{cases} \pi_k(\mathcal{M}) & k \leq n \\ 0 & \text{else}, \end{cases} \qquad (10)$$

and classify the homotopy defects of $\mathcal{M}$ using $\mathcal{M}_{D-1}$. $\mathcal{M}_n$ is called the $n$th Postnikov stage of $\mathcal{M}$ and obeys the fibration

$$B^n \pi_n(\mathcal{M}) \rightarrow \mathcal{M}_n \rightarrow \mathcal{M}_{n-1}, \qquad (11)$$

for $n \geq 2$. Each such fibration is classified by the twisted $(n+1)$ cocycle [97, 98]

$$[\beta^{n+1}] \in H_{\alpha_n}^{n+1}(\mathcal{M}_{n-1}, \pi_n(\mathcal{M})), \qquad (12)$$

called the Postnikov $(n+1)$-invariant. This makes the $n$th Postnikov stage $\mathcal{M}_n$ the classifying space of an $n$-group $\mathbb{G}_\pi^{(n)}$ (i.e., $\mathcal{M}_n = B\mathbb{G}_\pi^{(n)}$) which is defined by the data

$$\mathbb{G}_\pi^{(n)} = (\pi_1(\mathcal{M}) ; \pi_2(\mathcal{M}), \alpha_2, \beta^3 ; \cdots ; \pi_n(\mathcal{M}), \alpha_n, \beta^{n+1}). \qquad (13)$$

---

[4] See appendix A for a more detailed discussion.

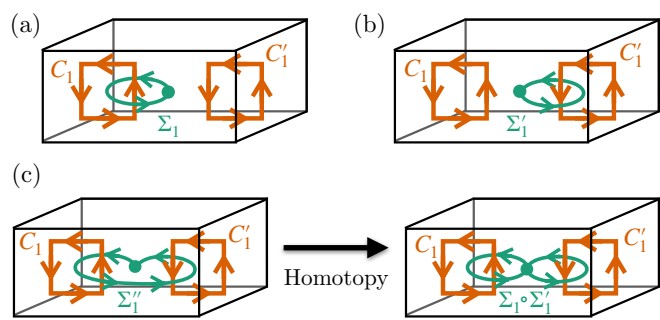

FIG. 1. A configuration $\mathcal{O}(x)$ in $D = 3$ with two homotopy defects $H(C_1)$ and $H(C_1')$ classified by $[S^1, \mathcal{M}]_{\mathrm{f}} = \mathbb{Z}$ is shown. (a) A defect $T_1(\Sigma_1)$ detects $H(C_1)$ and measures $[\mathcal{O}|_{\Sigma_1}]_{\mathrm{f}} = 1$ and (b) $T_1(\Sigma_1')$ detects $H(C_1')$ and measures $[\mathcal{O}|_{\Sigma_1'}]_{\mathrm{f}} = -1$. (c) The fusion $H(C_1) \times H(C_1')$ can be measured by $T_1(\Sigma_1'')$, but since $[\mathcal{O}|_{\Sigma_1''}]_{\mathrm{f}} = [\mathcal{O}|_{\Sigma_1 \circ \Sigma_1'}]_{\mathrm{f}}$ it can also be measured using $T_1(\Sigma_1 \circ \Sigma_1')$. This defines a multiplication structure $[\mathcal{O}|_{\Sigma_1 \circ \Sigma_1'}]_{\mathrm{f}} = [\mathcal{O}|_{\Sigma_1}]_{\mathrm{f}} \times [\mathcal{O}|_{\Sigma_1'}]_{\mathrm{f}}$, so $[\mathcal{O}|_{\Sigma_1''}]_{\mathrm{f}} = 0$ and $H(C_1) \times H(C_1')$ is the trivial homotopy defect line.

When all the $\alpha$ and $\beta$ are trivial, the classifying space of $\mathbb{G}_\pi^{(n)}$ is simply $B\mathbb{G}_\pi^{(n)} = \prod_{k=1}^n B^k \pi_k(\mathcal{M})$.

Because the homotopy defects of $\mathcal{M}$ in $D$-dimensional spacetime are the same as those from $\mathcal{M}_{D-1} = B\mathbb{G}_\pi^{(D-1)}$, the homotopy defects of $\mathcal{M}$ obey the same classification as magnetic defects (i.e., gauge fluxes) of $\mathbb{G}_\pi^{(D-1)}$ higher gauge theory [99]. When all Postnikov invariants $\beta^{n+1}$ are trivial for $n \leq D-1$, this reduces to the classification Eq. (7). Nontrivial Postnikov invariants reflect how lower codimension homotopy defects can carry the topological charge of higher codimension homotopy defects, which can be detected using $\Sigma_k \not\simeq S^k$.

The equivalence class $[\mathcal{O}|_{\Sigma_k}]_{\mathrm{f}}$ is measured by a defect $T_k(\Sigma_k)$ that detects $H(C_n)$ insertions for which the intersection number $\#(C_n, B_{k+1}) \neq 0$ (e.g., Fig. 1). The defect $T_k(\Sigma_k)$ is constructed using the topological invariant characterizing the free homotopy classes $[\mathcal{O}|_{\Sigma_k}]_{\mathrm{f}}$ (e.g., Eq. (25)). However, this topological invariant will generally not have a local expression in terms of $\mathcal{O}$.

## B. The higher gauge theory presentation

It is useful to consider a different presentation of the order parameter that makes the homotopy defects manifest. Consider instead $\widetilde{\mathcal{O}} : M_D \rightarrow \widetilde{G}$, where $\widetilde{G}$ is a covering space of $G$ with trivial $[\widetilde{\mathcal{O}}|_{\Sigma_k}]_{\mathrm{f}}$ classes for all $0 \leq k \leq D-1$ submanifolds. The original $G$ symmetry acting on $\mathcal{O}$ is realized as a $\widetilde{G}$ symmetry acting on $\widetilde{\mathcal{O}}$. As long as $\mathcal{M}$ admits a CW-decomposition, $\widetilde{G}$ for $\Sigma_k \cong S^k$ can always be constructed inductively using the Whitehead tower of $\mathcal{M}$ [100]. To have $\widetilde{\mathcal{O}}$-configurations correspond to $\mathcal{O}$-configurations, we turn on a 1-form $\widetilde{H}$ gauge field, where $\widetilde{H}$ is the cover of $H$ that lifts it to a subgroup of $\widetilde{G}$, to introduce the gauge

redundancy $\widetilde{\mathcal{O}}(x) \sim \widetilde{h}(x)\widetilde{\mathcal{O}}(x)$ with $\widetilde{h}(x) \in \widetilde{H}$. The gauge redundancy reduces the physically distinguishable values of $\widetilde{G}$ to $\widetilde{G}/\widetilde{H} = G/H \equiv \mathcal{M}$, and hence physical—gauge inequivalent—$\widetilde{\mathcal{O}}$-configurations correspond to $\mathcal{O}$-configurations.[5]

The main idea of this new presentation is that there are no $\widetilde{\mathcal{O}}$ homotopy defects, and all the information regarding the $\mathcal{O}$ homotopy defects are instead encoded in the $\widetilde{H}$ gauge fields. Indeed, because $\pi_n(\widetilde{G}) = 0$ for all $n \leq D - 1$, the long exact sequence of homotopy groups

$$\cdots \to \pi_2(\widetilde{H}) \to \pi_2(\widetilde{G}) \to \pi_2(\mathcal{M})$$
$$\to \pi_1(\widetilde{H}) \to \pi_1(\widetilde{G}) \to \pi_1(\mathcal{M}) \qquad (14)$$
$$\to \pi_0(\widetilde{H}) \to \pi_0(\widetilde{G}) \to \pi_0(\mathcal{M}),$$

yields the exact sequences

$$0 \to \pi_{n+1}(\mathcal{M}) \to \pi_n(\widetilde{H}) \to 0, \qquad n \leq D - 2, \quad (15)$$

and therefore

$$\pi_n(\widetilde{H}) = \pi_{n+1}(\mathcal{M}), \qquad n \leq D - 2. \qquad (16)$$

In this new presentation, codimension 2 homotopy defects are $\widetilde{H}$ gauge fluxes. However, codimension $> 2$ homotopy defects are encoded in the topology of the $\widetilde{H}$ gauge fields. When codimension $> 2$ homotopy defects exist, we can repeat the procedure by finding a covering space of $\widetilde{H}$ that trivializes this topology while preserving $\pi_0(\widetilde{H})$ and introducing a 2-form $K$ gauge field encoding these homotopy defects. From the long exact sequence of homotopy groups, this implies that $\pi_n(K) = \pi_{n+1}(\widetilde{H}) = \pi_{n+2}(\mathcal{M})$ for $0 \leq n \leq D - 3$, and thus codimension 3 homotopy defects are $K$ gauge fluxes. These 1-form and 2-form gauge fields may mix nontrivially, encoding how codimension 2 and 3 homotopy defects are related to and influence each other. This mixing is formally encoded by the total gauge group being a nontrivial 2-group.

This process can be repeated many times, and in the end, there will be a collection of gauge fields of varying forms, and the total gauge group will be the $(D-1)$-group $\mathbb{G}_\pi^{(D-1)}$. At the level of the homotopy defects, $\mathbb{G}_\pi^{(D-1)}$ will include all the data present in Eq. (13). However, it can also include additional structures reflecting the dynamics of the ordered phase as well. Importantly, like for Eq. (13), the homotopy defects are encoded directly as $\mathbb{G}_\pi^{(D-1)}$ gauge fluxes and not through the topology of target spaces, so the defect $T_k(\Sigma_k)$ will have a local expression in terms of the gauge fields.

As an example, consider $\mathcal{M} = \mathbb{RP}^2$ in $D = 3$ spacetime, where $\pi_0(\mathcal{M}) \simeq 0$, $\pi_1(\mathcal{M}) \simeq \mathbb{Z}_2$, and $\pi_2(\mathcal{M}) \simeq \mathbb{Z}$.

The action of $\pi_1(\mathcal{M})$ on $\pi_2(\mathcal{M})$ is described by the group homomorphism $\alpha_2 : \pi_1(\mathcal{M}) \to \mathrm{Aut}(\pi_2(\mathcal{M}))$ and changes the sign of $\pi_2(\mathcal{M})$. Thus, there are $\mathbb{Z}_2$ and $\mathbb{Z}_{\geq 0}$ codimension 2 and 3 homotopy defects, respectively. We first trivialize $\pi_1(\mathcal{M})$ using $\mathcal{O}' : M_3 \to S^2$ and a $\mathbb{Z}_2$ 1-form gauge field such that physical $\mathcal{O}'$ configurations take values in $S^2/\mathbb{Z}_2 \simeq \mathcal{M}$. Then, to trivialize $\pi_2(\mathcal{M})$ we lift $S^2$ to $\widetilde{G} \equiv S^3$, consider $\widetilde{\mathcal{O}} : M_3 \to S^3$, and introduce a $U(1)$ 1-form gauge field since $S^3/S^1 \simeq S^2$. The total gauge group is $\widetilde{H} = U(1) \times \mathbb{Z}_2$, which correctly recovers $S^3/(S^1 \times \mathbb{Z}_2) \simeq \mathbb{RP}^2 = \mathcal{M}$. The codimension 2 and 3 homotopy defects are the $\mathbb{Z}_2$ gauge fluxes and $U(1)$ magnetic monopoles, respectively. The latter is encoded by the topology of $U(1)$, so we lift $U(1)$ to $\mathbb{R}$ and introduce a 2-form $K \equiv \mathbb{Z}$ gauge field since $\mathbb{R}/\mathbb{Z} \simeq U(1)$. Including the group homomorphism $\alpha_2$, the homotopy defects are gauge fluxes of the discrete 2-group $\mathbb{G}_\pi^{(2)} = (\pi_1(\mathcal{M}), \pi_2(\mathcal{M}), \alpha_2)$ gauge theory, where we ignore the $\mathbb{R}$ 1-form gauge field since it has no magnetic defects. Indeed, triangulating spacetime and denoting the $\mathbb{G}_\pi^{(2)}$ gauge fields as the $\mathbb{Z}_2$ 1-cocycle $m_{ij}^{\mathbb{Z}_2}$ and $\mathbb{Z}$ valued 2-cochain $n_{ijk}^{\mathbb{Z}}$, the $\mathbb{G}_\pi^{(2)}$ gauge redundancy is[6]

$$m_{ij}^{\mathbb{Z}_2} \sim m_{ij}^{\mathbb{Z}_2} + (\mathrm{d}\lambda^{\mathbb{Z}_2})_{ij}, \qquad (17)$$

$$n_{ijk}^{\mathbb{Z}} \sim \alpha_2(\lambda_i^{\mathbb{Z}_2})n_{ijk}^{\mathbb{Z}} + (\mathrm{d}_{\alpha_2}\Lambda^{\mathbb{Z}})_{ijk}, \qquad (18)$$

where $\alpha_2$ enforces that the sign of the codimension 3 $\mathbb{Z}$ gauge fluxes is not physical.

## III. EMERGENT GENERALIZED SYMMETRIES

In the ordered phase, homotopy defects $H(C_{D-k-1})$ cost energy increasing with $|C_{D-k-1}|$ because the larger $|C_{D-k-1}|$ is, the more $\mathcal{O}$ deviates from $\mathcal{O}_{\mathrm{gs}}$. Thus, there is some low-energy scale $E < E_{\mathrm{IR}}$ without homotopy defects, and such configurations without homotopy defects can be globally continuously deformed to $\mathcal{O}_{\mathrm{gs}}$. In this regime, the defect $T_k(\Sigma_k)$ that detects $H(C_{D-k-1})$ is trivial for all $\Sigma_k$ since there are simply no homotopy defects. This also implies, in a trivial way, that $T_k(\Sigma_k)$ is a topological defect in this regime since it does not depend on the topology of $\Sigma_k$.

At $E > E_{\mathrm{IR}}$, configurations have homotopy defects $H(C_{D-k-1})$ inserted and $T_k(\Sigma_k)$ is no longer generally topological. Indeed, deforming $\Sigma_k$ such that the intersection number $\#(C_{D-k-1}, B_{k+1})$ changes will change the equivalence class $[\mathcal{O}|_{\Sigma_k}]_{\mathrm{f}}$.

In physical systems, homotopy defects generally come in two different types: they can be supported on closed submanifolds (i.e., $\partial C_{D-k-1} = \emptyset$) or can end on dynamical degrees of freedom residing along $\partial C_{D-k-1} \neq \emptyset$.

---

[5] This is a generalization of similar techniques that have been used in the study of magnets and nematics [101–103] and Villain formulations of lattice models [87, 104–107].

[6] See appendix A of Ref. 16 for a high-level introduction for physicists on twisted singular cohomology.

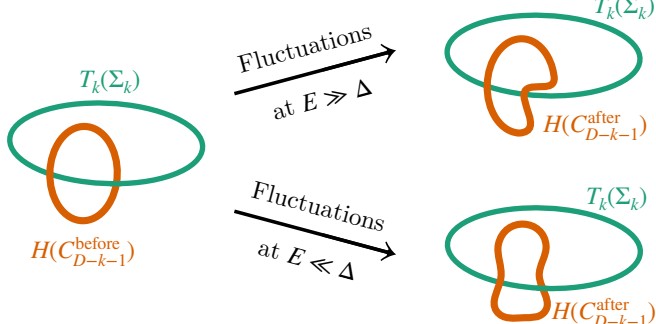

FIG. 2. The dynamics of homotopy defects depend on the energy scale. When $E \gg \Delta$, homotopy defects can end and $\text{link}(C_{D-k-1}^{\text{before}}, \Sigma_k) \neq \text{link}(C_{D-k-1}^{\text{after}}, \Sigma_k)$, so $\Sigma_k$ can become homologically trivial. However, when $E \ll \Delta$ and homotopy defects cannot end, $\text{link}(C_{D-k-1}^{\text{before}}, \Sigma_k) = \text{link}(C_{D-k-1}^{\text{after}}, \Sigma_k)$ is conserved, signaling the emergence of a symmetry.

From the perspective of "particle-vortex" duality, these degrees of freedom on $\partial C_{D-k-1}$ are the dynamical matter fields carrying gauge charge.

These dynamical degrees of freedom will have an energy gap $\Delta$. At energies $E_{\text{IR}} < E < \Delta$, the dynamical degrees of freedom are absent, and the only homotopy defects present are supported on closed submanifolds. In this regime, the only way to have $B_{k+1}$ no longer intersect $C_{D-k-1}$ is to deform $\Sigma_k$ through $C_{D-k-1}$. Therefore, $T_k(\Sigma_k)$ can be deformed as long as the linking number $\text{link}(\Sigma_k, C_{D-k-1})$ does not change. In other words, the shape of $\Sigma_k$ does not matter, only the linking number does [5, 6, 95]. This makes $T_k(\Sigma_k)$ a topological defect that generates a $(D - k - 1)$-form symmetry transforming $H(C_{D-k-1})$ when $\text{link}(\Sigma_k, C_{D-k-1}) \neq 0$. This was similarly discussed in Ref. 94 in the context of nonlinear sigma models.

We emphasize that this does not imply that the homotopy defects are topological defects, but only that they are *detected* by topological defects at low energy, which is a direct consequence of their homotopy-based classification reviewed in Sec. II A. In the $\widetilde{\mathcal{O}}$ presentation of $\mathcal{O}$ discussed in Sec. II B, $T_k(\Sigma_k)$ will be a topological quantum field theory on $\Sigma_k$ constructed from the $\mathbb{G}_\pi^{(D-1)}$ higher gauge fields. Furthermore, while our discussion thus far assumed that the spontaneously broken microscopic symmetry is invertible, the general principle applies to any generalized symmetries since their spontaneous breaking also gives rise to homotopy defects.

At $E > \Delta$, there are homotopy defects that can end, and the equivalence class $[\mathcal{O}|_{\Sigma_k}]_{\text{f}}$ can be changed by deforming $\Sigma_k$ such that $B_{k+1}$ goes through $\partial C_{D-k-1}$. Consequentially, the linking number $\text{link}(\Sigma_k, C_{D-k-1})$ loses its aforementioned topological properties. In fact, the linking number is no longer well defined for homotopy defects that do not end at $E > \Delta$ because they can now be "cut" open using endable homotopy defects, as shown in Fig. 2. Therefore, homotopy defects that can end and

screen the $(D - k - 1)$-form symmetry, explicitly breaking it.

When $\Delta \to \infty$, the $(D - k - 1)$-form symmetry is an exact symmetry. When $\Delta$ has a finite nonzero value, it is instead an emergent symmetry appearing at $E < \Delta$. This is an emergent higher-form symmetry for $k < D - 1$. Since the homotopy defect is the charged object under the symmetry, its symmetry charges are labeled by the homotopy defects' equivalence classes, and the symmetry sectors are the phase's topological sectors.

So, what is this symmetry? Let's restrict ourselves to the case where $\mathcal{M}$ is connected. Since the homotopy defects carry the symmetry charge, and since their classification is equivalent to that of magnetic defects of $\mathbb{G}_\pi^{(D-1)}$ higher gauge theory (see Eq. (13)), the symmetry is the same as the magnetic symmetry of $\mathbb{G}_\pi^{(D-1)}$ higher gauge theory. This means that the symmetry defects are the $\mathbb{G}_\pi^{(D-1)}$ electric defects (i.e., Wilson lines, surfaces, etc.) and described by the $(D - 1)$-representations of $\mathbb{G}_\pi^{(D-1)}$. When there are finitely many homotopy defect classes, $\mathbb{G}_\pi^{(D-1)}$ is finite, and the symmetry category[7] $\mathcal{S}$ describing this emergent symmetry is the fusion $(D - 1)$-category

$$\mathcal{S} = (D - 1)\text{-Rep}(\mathbb{G}_\pi^{(D-1)}). \tag{19}$$

This symmetry includes non-invertible and invertible symmetries as well as 0-form and higher-form symmetries. However, it does not include any information about codimension 1 homotopy defects whose classification is related to $\pi_0(\mathcal{M})$. At the most basic level, Eq. (19) is the emergent symmetry when restricted to a single superselection section in the ground state subspace of the SSB phase. However, more generally, the $(D - 1)$-form symmetry arising from codimension 1 homotopy defects can mix nontrivially. The emergent symmetry associated with codimension 1 homotopy defects is additional information, which we will discuss in Sec. III D.

This expression for $\mathcal{S}$ can be directly verified using a $(D + 1)$-dimensional topological field theory (TFT) called the Symmetry TFT (SymTFT)[8] [22, 23, 71, 108–126]. Its topological defects are described by the Drinfeld center $\mathcal{Z}(\mathcal{S})$ of $\mathcal{S}$. A defining feature of the SymTFT is that it has a topological boundary condition $\mathfrak{B}_{\text{sym}}$ whose topological defects are described by $\mathcal{S}$. The topological defects in $\mathcal{Z}(\mathcal{S})$ that become the trivial defect on $\mathfrak{B}_{\text{sym}}$ correspond to the symmetry charges of $\mathcal{S}$. Therefore, one can find $\mathcal{S}$ from its symmetry charges by: (1) constructing a TFT with topological defects corresponding to symmetry charges; (2) finding a topological boundary where they become trivial; (3) finding the fusion higher

---

[7] See appendix B for an introduction on symmetry categories.
[8] See appendix C for an introduction on the SymTFT. We note that the SymTFT has also been called categorical symmetry [22, 108–110], holographic categorical symmetry [111], topological holography [112], topological symmetry [113], and symmetry topological order (SymmTO) [114, 115].

category describing nontrivial topological defects on that boundary, which will be $\mathcal{S}$.

Since the emergent symmetries are exact symmetries of finite $\mathbb{G}_\pi^{(D-1)}$ gauge theory in $D$ dimensional spacetime, the natural candidate for the SymTFT is $\mathbb{G}_\pi^{(D-1)}$ gauge theory in one higher dimension, whose topological defects are described by $\mathcal{Z}((D-1)\text{-Vec}_{\mathbb{G}_\pi^{(D-1)}})$. From standard dimensional reduction [127], it is clear that the SymTFT's gauge fluxes correspond to $\mathbb{G}_\pi^{(D-1)}$ gauge fluxes in $D$-dimensions and, therefore, the homotopy defects. The SymTFT has two canonical topological boundaries: an electric and magnetic boundary whose topological defects are described by $(D-1)\text{-Vec}_{\mathbb{G}_\pi^{(D-1)}}$ and $(D-1)\text{-Rep}(\mathbb{G}_\pi^{(D-1)})$, respectively. On the electric (magnetic) boundary, the electric defects (gauge fluxes) become the trivial defect. Since the gauge fluxes are trivial on the magnetic boundary and they correspond to the symmetry charge, $\mathfrak{B}_{\text{sym}}$ is the magnetic boundary, and the symmetry category is indeed Eq. (19).

This expression for $\mathcal{S}$ is abstract and hides a lot of physical insights. Therefore, we will now focus on better understanding the emergent symmetry and the SymTFT construction by working through simple examples.

## A. Invertible homotopy defects

Let us first consider the case where all homotopy defects are invertible. By this, we mean that they are classified by $[S^k, \mathcal{M}]_{\text{f}} \simeq \pi_k(\mathcal{M})$ and their fusion is described by the groups $\pi_k(\mathcal{M})$, which are necessarily abelian. Then, $\mathbb{G}_\pi^{(D-1)} = \prod_{r=0}^{D-2} G^{(r)}$ is a trivial higher group and just a direct product of $G^{(r)} = \pi_{r+1}(\mathcal{M})$. The symmetry category (19) becomes

$$\mathcal{S} = (D-1)\text{-Vec}_{\hat{\mathbb{G}}_\pi^{(D-1)}}, \qquad (20)$$

where $\hat{\mathbb{G}}_\pi^{(D-1)} = \prod_{r=0}^{D-2} \hat{G}^{(r)}$ is a trivial $(D-1)$-group with $\hat{G}^{(r)} = \text{Hom}(\pi_{D-r-1}(\mathcal{M}), U(1))$ the Pontryagin dual of $\pi_{D-r-1}(\mathcal{M})$. This symmetry category describes finite $r$-form $\hat{G}^{(r)}$ symmetries and their condensation defects [128].

We can understand this symmetry more transparently, and also for the continuous case, using that the homotopy groups describe its symmetry charges. Since they are abelian, they are direct products of $\mathbb{Z}$ and $\mathbb{Z}_N$ in physically relevant phases. So, the $(D-k-1)$-form symmetry's charges are labeled by $\mathbb{Z}$ and $\mathbb{Z}_N$, and the corresponding symmetry groups are $U(1)$ and $\mathbb{Z}_N$, respectively. In general, as in Eq. (20), the $(D-k-1)$-form symmetry is invertible and described by $\text{Hom}(\pi_k(\mathcal{M}), U(1))$. Therefore, if codimension $k+1$ invertible homotopy defects are classified by $\mathbb{Z}_N^n \times \mathbb{Z}^m$, there is an emergent $\mathbb{Z}_N^n \times U(1)^m$ $(D-k-1)$-form symmetry.

Many ordered phases have SSB patterns that give rise to invertible homotopy defects. A simple example is an isotropic magnet where $SO(3) \xrightarrow{\text{ssb}} SO(2)$. The 1st homotopy group of $\mathcal{M} \simeq S^2$ is trivial while $\pi_2(\mathcal{M}) = \mathbb{Z}$, so there is an emergent $U(1)^{(D-3)}$ symmetry. A related example is a noncollinear anti-ferromagnet where $SO(3) \xrightarrow{\text{ssb}} 1$. It has codimension 2 homotopy defects classified by $\pi_1(SO(3)) \simeq \mathbb{Z}_2$ and thus an emergent $\mathbb{Z}_2^{(D-2)}$ symmetry.

Let us demonstrate this explicitly in a simple case. Consider the partition function of a non-linear sigma model

$$Z(M_D) = \int \mathcal{D}[\mathcal{O}] \, \text{e}^{-S(\mathcal{O}, M_D)}, \qquad (21)$$

where, from the coset construction, $\mathcal{O} : M_D \to \mathcal{M}$ is the Goldstone field and $S$ is the general effective Euclidean action [129–132]. For codimension $k+1$ homotopy defects classified by $\pi_k(\mathcal{M}) = \mathbb{Z}$, the topological charge $[\mathcal{O}|_{\Sigma_k}]_{\text{f}} \in \mathbb{Z}$ can be expressed as $Q^{\text{top}}[\Sigma_k] = \int_{\Sigma_k} * J^{\text{top}}$ using the $(D-k)$ form current $J^{\text{top}}$ [133].[9] In the absence of homotopy defects, $* J^{\text{top}}$ is closed and is the generator of the de Rham cohomology of $\mathcal{M}$ pulled back to $M_D$ using $\mathcal{O}$ [134].

We can reveal the emergent symmetry explicitly by constraining the partition function to only integrate over fields satisfying $\text{d} * J^{\text{top}} = 0$. Enforcing this using a $(D-k-1)$-form Lagrange multiplier field $\lambda$, the partition function becomes

$$Z_{\text{eff}}(M_d) = \int \mathcal{D}[\mathcal{O}] \mathcal{D}[\lambda] \text{e}^{-S(\mathcal{O}, M_D) - \text{i} \int_{M_D} \text{d}\lambda \wedge * J^{\text{top}}}. \qquad (22)$$

Since $Q^{\text{top}} \in \mathbb{Z}$, the $\lambda$ is a $(D-k-1)$-form $U(1)$ gauge field.

The partition function $Z_{\text{eff}}$ has a new symmetry absent from $Z$: it is invariant under the transformation

$$\lambda \to \lambda + \Gamma, \qquad \text{d}\Gamma = 0. \qquad (23)$$

Turning on a $(D-k)$-form background field $\mathcal{A}$ for this symmetry introduces the gauge redundancy

$$\lambda \to \lambda + \Gamma, \qquad \mathcal{A} \to \mathcal{A} + \text{d}\Gamma. \qquad (24)$$

Minimally coupling $\mathcal{A}$, we replace $\text{d}\lambda$ in Eq. (22) by $\text{d}\lambda - \mathcal{A}$. This introduces the term $\text{i}\mathcal{A} \wedge * J^{\text{top}}$ into the effective action, revealing that the Noether current of this symmetry is $J^{\text{top}}$. Thus, the generator of the symmetry is

$$T_k^{(\alpha)}(\Sigma_k) = \text{e}^{\text{i}\alpha \int_{\Sigma_k} * J^{\text{top}}}, \qquad (25)$$

and when $\Sigma_k$ and $C_{D-k-1}$ link it transforms the homotopy defect $H(C_{D-k-1}) = \text{e}^{\text{i} \int_{C_{D-k-1}} \lambda}$ by $\text{e}^{\text{i}\alpha}$. Since $Q^{\text{top}} \in \mathbb{Z}$, the parameter $\alpha \in [0, 2\pi)$ and $T_k^{(\alpha)}$ generates a $U(1)^{(D-k-1)}$ symmetry.

───────

[9] The simplest example is $* J^{\text{top}} = \text{d}\theta$ where $\theta : M_D \to \mathbb{R}/\mathbb{Z}$. More generally, $J^{\text{top}}$ is a properly normalized topological term constructed from the Maurer-Cartan form $\mathcal{O}^{-1}\text{d}\mathcal{O}$.

## B. Higher-form symmetry homotopy defects

It is straightforward to generalize our discussion on invertible homotopy defects of 0-form invertible symmetries to those arising from spontaneously breaking an invertible higher-form symmetry $G^{(p)} \xrightarrow{\text{ssb}} H^{(p)}$. As mentioned in footnote 3, they are classified by the target space $\mathcal{M} = B^p(G/H)$, and since $G$ is always an abelian group, they are always invertible. We'll consider two simple cases before stating the general result.

For the SSB pattern $\mathbb{Z}_N^{(p)} \xrightarrow{\text{ssb}} \mathbb{Z}_M^{(p)}$, where $N/M$ is a positive integer, the order parameter manifold is $\mathcal{M} = B^p \mathbb{Z}_{N/M}$ and homotopy defects are classified by

$$\pi_k(B^p \mathbb{Z}_{N/M}) = \pi_{k-p}(\mathbb{Z}_{N/M}) = \begin{cases} \mathbb{Z}_{N/M} & k = p, \\ 0 & k \neq p. \end{cases} \quad (26)$$

Therefore, there are codimension $(p+1)$ $\mathbb{Z}_N$ homotopy defects in this SSB phase. Since these are the emergent symmetry's charges, it is a $\mathbb{Z}_{N/M}^{(D-p-1)}$ symmetry. We remark that when $M = 1$, this SSB pattern is realized in $p$-form $BF$ theory (i.e., in ground states of $p$-form toric code). In this theory, the homotopy defects are the gauge fluxes, and the $\mathbb{Z}_N^{(D-p-1)}$ symmetry is the magnetic symmetry [135].

For the SSB pattern $U(1)^{(p)} \xrightarrow{\text{ssb}} \mathbb{Z}_M^{(p)}$, the order parameter manifold is $\mathcal{M} = B^p S^1$, and the homotopy defects are classified by

$$\pi_k(B^p S^1) = \pi_{k-p}(S^1) = \begin{cases} \mathbb{Z} & k = p+1, \\ 0 & k \neq p+1. \end{cases} \quad (27)$$

Therefore, there are codimension $(p+2)$ $\mathbb{Z}$ homotopy defects in this SSB phase and, when they cannot end, an emergent $U(1)^{(D-p-2)}$ symmetry. When $M = 1$, this SSB pattern is realized in $p$-form Maxwell theory. The homotopy defects are the magnetic world volumes (e.g., 't Hooft lines), and the $U(1)^{(D-p-2)}$ symmetry is the magnetic symmetry [7].

To write down a general expression for the emergent symmetry, we can use the $p$-loop space $\Omega^p \mathcal{M}$ of $\mathcal{M}$. For the two SSB patterns considered above, at the level of homotopy $\Omega^p \mathcal{M} \simeq \mathbb{Z}_{N/M}$ and $\Omega^p \mathcal{M} \simeq S^1$, respectively. For a general $G^{(p)} \xrightarrow{\text{ssb}} H^{(p)}$ SSB pattern, the order parameter manifold will satisfy $\Omega^p \mathcal{M} \simeq G/H$ and each emergent $(D-k-p-1)$-form symmetry will be described by the Pontryagin dual of $\pi_k(\Omega^p \mathcal{M})$.

## C. Codimension 2 homotopy defects

We next consider codimension 2 homotopy defects. They are classified by the conjugacy classes $\text{Cl}(\pi_1(\mathcal{M}))$ of $\pi_1(\mathcal{M})$. When $\pi_1(\mathcal{M})$ is abelian, $\text{Cl}(\pi_1(\mathcal{M})) = \pi_1(\mathcal{M})$ and the discussion from Sec. III A applies. Here we will consider general finite $\pi_1(\mathcal{M})$, which can be abelian or

non-abelian. From the previous general discussion, when these homotopy defects cannot end, there is a $(D-2)$-form symmetry whose symmetry charges are labeled by $\text{Cl}(\pi_1(\mathcal{M}))$. The symmetry category is given by Eq. (19) with only the codimension 2 homotopy defects included. Therefore, it is

$$\mathcal{S} = (D-1)\text{-Rep}(\pi_1(\mathcal{M})), \quad (28)$$

which describes a $\text{Rep}(\pi_1(\mathcal{M}))$ $(D-2)$-form symmetry and its condensation defects. In Sec. IV A, we consider a general model in $D = 3$ that has codimension 2 homotopy defects for which we find an expression for the topological defect generating the $\text{Rep}(\pi_1(\mathcal{M}))$ 1-form symmetry.

There are numerous ordered phases whose SSB patterns give rise to codimension 2 homotopy defects. Since we already discussed examples of invertible codimension 2 homotopy defects in III A, let us consider examples where $\pi_1(\mathcal{M})$ is non-abelian. A generic example is an Eilenberg–MacLane space $\mathcal{M} = K(F, 1)$ of a finite non-abelian group $F$. The only nontrivial homotopy group is $\pi_1(\mathcal{M}) = F$, and the emergent symmetry is $(D-1)\text{-Rep}(F)$. A more physical example is an ordered phase with the SSB pattern $SO(3) \xrightarrow{\text{ssb}} \mathbb{Z}_2 \times \mathbb{Z}_2$, which occurs in biaxial nematic liquid crystals [136] and certain spin-1 quantum magnets [103, 137]. In this ordered phase, $\pi_1(\mathcal{M}) = Q_8$, where $Q_8$ is the Quaternion group. Therefore, when the codimension-2 homotopy defects in this spin-1 magnetic phase cannot end, there is an emergent $(D-1)\text{-Rep}(Q_8)$ symmetry, which includes a $\text{Rep}(Q_8)$ $(D-2)$-form symmetry.

As shown in the beginning of this section, the symmetry category can also be deduced using the SymTFT. Let us consider the construction in greater detail for this simple case. We first set $D = 2$ and consider the $(2+1)$-dimensional quantum double model $\mathbf{D}(\pi_1(\mathcal{M}))$. Its topological defect lines—its anyons—are the objects $(\text{Cl}(k), \alpha_k)$ of $\mathcal{Z}(\text{Vec}_{\pi_1(\mathcal{M})})$, where $\text{Cl}(k)$ is the conjugacy class of $k \in \pi_1(\mathcal{M})$ and $\alpha_k$ an irreducible representation of the centralizer $C(k) = \{a \in \pi_1(\mathcal{M}) \mid ak = ka\}$ of $k$ [138]. $\mathbf{D}(\pi_1(\mathcal{M}))$ has a topological boundary $\mathscr{B}$ where the $(\text{Cl}(k), \mathbf{1})$ topological defect lines can end [139]. Therefore, treating $\mathbf{D}(\pi_1(\mathcal{M}))$ as a symTFT, the topological boundary $\mathscr{B}$ describes a symmetry whose charges are labeled by $\text{Cl}(\pi_1(\mathcal{M}))$, which is precisely what we are seeking.

The symmetry category $\mathcal{S}$ is, therefore, the fusion category describing the topological defects on the boundary $\mathscr{B}$. We can find $\mathcal{S}$ using the correspondence [22, 112, 114, 139–141] between topological boundaries of $\mathbf{D}(K)$ and gapped phases of $(1+1)$-dimensional theories with a $K$ symmetry [142, 143]. The boundary $\mathscr{B}$ corresponds to a trivial $K$ SPT phase and its gapped excitations are $K$ charges, which are described by $\text{Rep}(K)$ [22]. Heuristically, $(\text{Cl}(k), \mathbf{1})$ topological defects become the trivial topological defect on $\mathscr{B}$, so a general topological defect $(\text{Cl}(k), \alpha_k)$ on $\mathscr{B}$ is described by $\text{Rep}(K)$ since it "forgets" its $\text{Cl}(k)$ label. Therefore, setting $K = \pi_1(\mathcal{M})$, the

topological defects on $\mathscr{B}$ are described by $\mathsf{Rep}(\pi_1(\mathcal{M}))$ and the symmetry category is $\mathcal{S} = \mathsf{Rep}(\pi_1(\mathcal{M}))$.

For general $D$, we consider $D+1$ dimensional pure $\pi_1(\mathcal{M})$ gauge theory as the symTFT, which has topological defects labeled by the objects in $\mathcal{Z}((D-1)\text{-}\mathsf{Vec}_{\pi_1(\mathcal{M})})$. Using that

$$\mathcal{Z}((D-1)\text{-}\mathsf{Vec}_K) = \mathcal{Z}((D-1)\text{-}\mathsf{Rep}(K)), \qquad (29)$$

there is a topological boundary whose topological defects are labeled by $(D-1)\text{-}\mathsf{Rep}(\pi_1(\mathcal{M}))$. This is also the boundary where gauge fluxes, codimension 2 topological defects labeled by $\mathrm{Cl}(\pi_1(\mathcal{M}))$, can end [22]. Therefore, the symmetry category is $\mathcal{S} = (D-1)\text{-}\mathsf{Rep}(\pi_1(\mathcal{M}))$, in agreement with Eq. (28).

### D. Codimension 1 homotopy defects

As mentioned, the symmetry category Eq. (19) does not include symmetries arising from codimension 1 homotopy defects (i.e., domain walls), which are classified by $\pi_0(\mathcal{M})$. Here we will explore these emergent symmetries and, for simplicity, restrict ourselves to the symmetry-breaking pattern $G \xrightarrow{\text{ssb}} H$ where $G$ is a finite group and $H$ is a normal subgroup. The SSB phase is then gapped, and the homotopy defects are all codimension 1. They are measured by comparing the order parameter $\mathcal{O}$ at two points in $M_D$. Therefore, in the absence of domain walls—in the ground state subspace—$\mathcal{O}$ becomes a topological defect that generates a $(D-1)$-form symmetry.

In this case, $\pi_0(\mathcal{M}) \simeq \mathcal{M}$ is a finite group that classifies invertible codimension 1 homotopy defects and their fusion. Furthermore, the local order parameters that acquire a vev are labeled by and fuse according to $\mathsf{Rep}(\mathcal{M})$. Since they are the topological defects generating the symmetry, we find that they generate a $\mathsf{Rep}(\mathcal{M})$ $(D-1)$-form symmetry. This is a non-invertible symmetry when $\mathcal{M}$ is non-abelian, and when $\mathcal{M}$ is abelian, it is an invertible described by the Pontryagin dual of $\mathcal{M}$.

We can also deduce this using the SymTFT. Since $\pi_0(\mathcal{M}) = \mathcal{M}$, the charged objects are codimension 1 and labeled by group elements of $\mathcal{M}$. We consider the $(D+1)$ SymTFT that describes the ground states of a $\mathcal{M} \xrightarrow{\text{ssb}} 1$ SSB phase. It has local topological defects labeled by $\mathsf{Rep}(\mathcal{M})$ and codimension 1 topological defects labeled by group elements of $\mathcal{M}$. $\mathfrak{B}_{\text{sym}}$ is the boundary where this codimension 1 topological defect is trivial. The topological defects of $\mathfrak{B}_{\text{sym}}$ are then the local topological defects, and therefore the symmetry is a $\mathsf{Rep}(\mathcal{M})$ $(D-1)$-form symmetry.

For example, consider a $Z_N$ spin ferromagnetic phase with the SSB pattern $\mathbb{Z}_N \xrightarrow{\text{ssb}} \mathbb{Z}_M$, where $N/M$ is a positive integer. The order parameter manifold is $\mathcal{M} = \mathbb{Z}_{N/M}$, which is self-Pontryagin dual, and so there is an emergent $\mathbb{Z}_{N/M}^{(D-1)}$ symmetry in the ground state subspace. $G$ can also be non-abelian. For example, consider $G = S_3$, the symmetric group of degree 3. For the

SSB pattern $S_3 \xrightarrow{\text{ssb}} \mathbb{Z}_3$, the order parameter manifold is $\mathcal{M} = \mathbb{Z}_2$, so there is an emergent $\mathbb{Z}_2^{(D-1)}$ symmetry. On the other hand, for $S_3 \xrightarrow{\text{ssb}} 1$, the order parameter manifold is $\mathcal{M} = S_3$ and so there is an emergent $\mathsf{Rep}(S_3)$ $(D-1)$-form symmetry.

Since the local topological defect $\mathcal{O}$ is the order parameter, it interacts nontrivially with the codimension 1 topological defect $T_{D-1}^{(g)}$ generating the $G$ 0-form symmetry. Indeed, when $\mathcal{O}$ transforms in the representation $R \in \mathsf{Rep}(G)$,

$$T_{D-1}^{(g)}(\Sigma)\mathcal{O}_i(x) = \begin{cases} R_{ij}^{(g)}\mathcal{O}_j(x) & \mathrm{link}(\Sigma, x) \neq 0, \\ \mathcal{O}_i(x) & \mathrm{link}(\Sigma, x) = 0, \end{cases} \quad (30)$$

where $g \in G$. This defines an action of the $G$ 0-form symmetry on the emergent $(D-1)$-form symmetry. When the emergent symmetry is invertible, this action is encoded by the group homomorphism

$$\rho : G \to \mathrm{Aut}(\mathcal{M}), \qquad (31)$$

and the total symmetry is a split $D$-group $\mathbb{G}^{(D)} = (G, \mathcal{M}, \rho)$.

## IV. DISORDERING

As shown in Sec. III, when a 0-form invertible symmetry $G$ is spontaneously broken, there is an emergent generalized symmetry $\mathcal{S}$ at low energies. However, this emergent symmetry is *not* spontaneously broken in the ordered phase.[10] A cheeky reason why is that if it were, then ordinary ordered phases would have topological orders and emergent photons, which is certainly not true. The better explanation comes from a hallmark feature of homotopy defects in ordered phases: their energy cost grows as they are separated/enlarged in space—they are confined. Therefore, the homotopy defects obey an "area law" at low energies in the ordered phase and appear as gapped extended objects in the spectrum. So, the emergent symmetry $\mathcal{S}$ is not spontaneously broken.

While $\mathcal{S}$ is not spontaneously broken when it emerges in the ordered phase, nothing generally prevents it from becoming spontaneously broken. Of course, it is not always possible for parts of $\mathcal{S}$ to spontaneously break due to generalized Mermin-Wagner-Coleman theorems. An invertible finite (continuous) $p$-form symmetry can only spontaneously break when $D > p + 1$ ($D > p + 2$) [7, 38]. We expect this criterion also to apply when the symmetry is non-invertible. Therefore, the emergent symmetry associated with finite $[\Sigma_k, \mathcal{M}]_{\mathrm{f}}$ can only spontaneously break if $k \geq 1$, and for nonfinite $[\Sigma_k, \mathcal{M}]_{\mathrm{f}}$ if $k \geq 2$.

―――――

[10] For higher-form symmetries discussed in Sec. III B, we expect the emergent symmetry to be spontaneously broken due to a mixed 't Hooft anomaly [68].

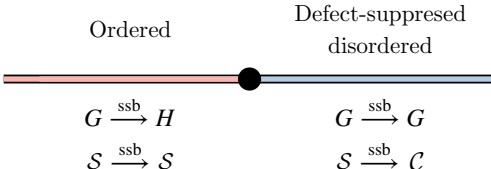

FIG. 3. Spontaneously breaking the emergent symmetry $\mathcal{S}$ to a subcategory $\mathcal{C}$ restores $G$ and drives a phase transition to a non-trivial disordered phase. The ordered and defect-disordered phases have district SSB patterns, making the critical point between them a possible generalized DQCP. When $\mathcal{S}$ is an invertible 0-form symmetry, the critical point is an ordinary DQCP.

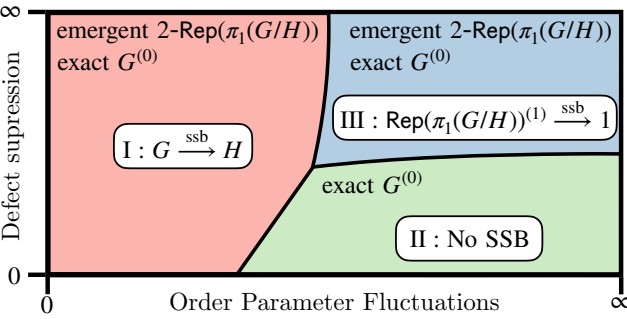

FIG. 4. Shows the proposed phase diagram in $D = 3$ of an (I) ordered phase with codimension 2 homotopy defects and its (II) trivial and (III) defect-suppressed disordered phases, labeled by their symmetry breaking patterns. The vertical axis shows the suppression of the homotopy defects, while the horizontal axis controls the fluctuations of the local order parameter. When the defect suppression goes to infinity, the 2-Rep$(\pi_1(G/H))$ symmetry is exact. Using the gauge theoretic language of the $\widetilde{\mathcal{O}}$ presentation (see Sec. II B), regions I, II, and III are the Higgs, confined, and deconfined phases, respectively, of $\mathbb{G}_\pi^{(D-1)}$ gauge theory.

Undergoing a phase transition that spontaneously breaks $\mathcal{S}$ causes the homotopy defects to obey a "perimeter law" at low energies, which signals that the homotopy defects are deconfined. Furthermore, in Lorentzian signature, where we call something a defect if it extends in the time direction and an operator acting on the Hilbert space, the excitations created by homotopy operators will form a condensate since they carry $\mathcal{S}$ symmetry charge. These features completely contradict the aforementioned ones of the $G \xrightarrow{\text{ssb}} H$ phase. Therefore, it is not possible to spontaneously break both $G$ and $\mathcal{S}$ simultaneously, and spontaneously breaking $\mathcal{S}$ must drive a transition to a disordered phase.

The $\mathcal{S}$ SSB phase will generally have topological order and emergent photons and will be enriched by the microscopic $G$ symmetry [144, 145]. Indeed, for finite $\mathcal{S}$, it will correspond to the deconfined phase of $\mathbb{G}_\pi^{(D-1)}$ gauge theory. Furthermore, spontaneously breaking $\mathcal{S}$ will give rise to new homotopy defects, and thus, there will be new emergent symmetries in the $\mathcal{S}$ SSB phase at low energies.

Since spontaneous breaking $\mathcal{S}$ restores the $G$ symmetry, it induces a direct transition between two different SSB phases. If the transition is continuous order and $\mathcal{S}$ is an invertible 0-form symmetry, the critical point is a deconfined quantum critical point (DQCP) [146]. For example, Ref. 147 studied a transition between an $SO(3) \xrightarrow{\text{ssb}} U(1)$ phase in $D = 3$, where $\mathcal{S}$ describes a $U(1)^{(0)}$ symmetry, and the $\mathcal{S}$ SSB phase, and the critical theory is the same as the original DQCP proposed by Ref. 148. When $\mathcal{S}$ is a generalized symmetry, the critical point would then be a generalized DQCP (see Fig. 3). Models with generalized DQCPs can be constructed by taking a model with an ordinary continuous SSB transition and then gauging a finite subgroup of the symmetry that controls the transition [149].

We expect it is impossible to have both $G$ and $\mathcal{S}$ simultaneously not spontaneously broken. Since $\mathcal{S}$ emerges because of homotopy defects arising from spontaneously breaking $G \xrightarrow{\text{ssb}} H$, heuristically, restoring $G$ should cause a qualitative change to the homotopy defects and also the realization of $\mathcal{S}$. If the conjecture made in Sec. V is correct, this could be due to a mixed 't Hooft anomaly between $G$ and $\mathcal{S}$. Therefore, starting in the ordered phase and restoring $G$, $\mathcal{S}$ must either spontaneously break or no longer emerge, leading to nontrivial or trivial disordered phases, respectively.

The easiest way to spontaneously break $\mathcal{S}$ is to disorder while suppressing homotopy defects, preventing them from proliferating and destroying $\mathcal{S}$. When this defect suppression is infinity and $\mathcal{S}$ is an exact symmetry (e.g., Eq. (22)), the trivial disordered phase is inaccessible, and any disordered phase will spontaneously break $\mathcal{S}$. When the defect suppression is large but finite, the $\mathcal{S}$ SSB phase will persist only if $\mathcal{S}$ is a higher-form symmetry because emergent higher-form symmetries are exact [88] (see Fig. 4).

Defect-suppressed disordered phases have been studied previously in particular models [101–103, 137, 147, 150–153]. Here we interpret these examples in terms of the emergent symmetry $\mathcal{S}$ being spontaneously broken. This offers a general, unifying framework that can be applied predictively to ordered phases in any dimension, regardless of how complicated their homotopy defects are.

The emergent symmetry can spontaneously break in other ways. A particularly interesting scenario is when it is a continuous symmetry and has conserved currents from which its generators are constructed [7, 154]. Taking the wedge product of these currents yields a new conserved composite current corresponding to a new symmetry that transforms topologically linked homotopy defects [92]. Therefore, condensing topologically linked homotopy defects spontaneously breaks part of the emergent continuous symmetry and provides an alternative route to a nontrivial disordered phase. As an example, consider a two-component superfluid in $D = 4$, where $U(1) \times U(1) \xrightarrow{\text{ssb}} 1$ and there

are Goldstone fields $\theta_1, \theta_2 \in \mathbb{R}/\mathbb{Z}$. In the deep IR, there are two emergent $U(1)^{(2)}$ symmetries with currents $*j_{1,2} = \mathrm{d}\theta_{1,2}$. Additionally, there is an emergent $U(1)^{(1)}$ symmetry whose conserved current is the composite current $*j = \mathrm{d}\theta_1 \wedge \mathrm{d}\theta_2$. While the 2-form symmetries cannot spontaneously break, the 1-form symmetry can, giving rise to a nontrivial disordered phase with emergent photons. A microscopic mechanism for spontaneously breaking this 1-form symmetry was discussed in Ref. 155.

In the remainder of this section, we contextualize this abstract discussion and explore examples of defect-suppressed disordered phases using the $\widetilde{\mathcal{O}}$ presentation of the order parameter and simple Euclidean lattice models. We build the spacetime lattice $\mathbb{M}_D$ by triangulating $M_D$ and will denote 0-simplices (lattice sites) by $i$, 1-simplices (edges) that connect $i$ and $j$ as $(ij)$, 3-simplices (triangles) whose corners are $i$, $j$, and $k$ as $(ijk)$, etc.

### A. Suppressing codimension 2 homotopy defects

The simplest example where the emergent symmetry $\mathcal{S}$ can spontaneously break is in $D = 3$ when the symmetry breaking pattern $G \xrightarrow{\text{ssb}} H$ has an order parameter manifold $\mathcal{M} \equiv G/H$ with finitely many classes of codimension two homotopy defects. In this case, since $\pi_1(\mathcal{M})$ is finite, the emergent symmetry is $\mathcal{S} = 2\text{-}\mathsf{Rep}(\pi_1(\mathcal{M}))$ (see Sec. III C). For simplicity, we will assume that $H$ is a finite subgroup of $G$ and that $G$ is connected.

To describe these homotopy defects' dynamics, we use the $\widetilde{\mathcal{O}}_i \in \widetilde{G}$ presentation of the order parameter $\mathcal{O}_i \in \mathcal{M}$ discussed in Sec. II B. Since we require $\widetilde{\mathcal{O}}$ to have no codimension 2 homotopy defects, we choose $\widetilde{G}$ to be the universal cover of $G$ as it satisfies $\pi_1(\widetilde{G}) = 0$. So, on each 0-simplex $i$ resides a $\widetilde{G}$ degree of freedom $\widetilde{\mathcal{O}}_i$ and on each 1-simplex $(ij)$ resides an $\widetilde{H}$ gauge field $a_{ij} \equiv a_{ji}^{-1}$, where $\widetilde{H}$ is the cover of $H$ the lifts it to a subgroup of $\widetilde{G}$. The $\widetilde{H}$ gauge redundancy

$$\widetilde{\mathcal{O}}_i \sim \widetilde{h}_i \widetilde{\mathcal{O}}_i, \qquad a_{ij} \sim \widetilde{h}_i a_{ij} \widetilde{h}_j^\dagger, \qquad (32)$$

with gauge parameter $\widetilde{h}_i \in \widetilde{H}$, enforces physical $\widetilde{\mathcal{O}}$ configurations to correspond to $\mathcal{O}$ configurations.

The homotopy defects are strings in spacetime, residing on the 1-simplices of the dual lattice, which are equivalently the 2-simplices of the direct lattice. They are detected by $\mathcal{O}$'s homotopy class along a loop, which in the $\widetilde{\mathcal{O}}$ presentation is probed by acting $a_{ij}$ along a loop. Therefore, the homotopy defects are violations of the flatness condition $(\delta a)_{ijk} = a_{ij} a_{jk} a_{ki} = 1$ and correspond to $\widetilde{H}$ gauge fluxes, which are classified by $\mathrm{Cl}(\widetilde{H})$ [156].

We can directly verify this from the exact sequence Eq. (15). Because $\widetilde{H}$ is finite here, it provides the exact sequence

$$0 \to \pi_1(\mathcal{M}) \to \widetilde{H} \to 0. \qquad (33)$$

Therefore, $\pi_1(\mathcal{M}) \simeq \widetilde{H}$ and codimension 2 homotopy defects are indeed classified by $\mathrm{Cl}(\widetilde{H})$.

When the homotopy defects are gapped and cannot end, the ground state always satisfies $(\delta a)_{ijk} = a_{ij} a_{jk} a_{ki} = 1$ on all 2-simplices. Therefore the Wilson loop

$$T_1(\gamma) = \mathrm{Tr} \prod_{(ij) \in \gamma} a_{ij}, \qquad (34)$$

where $\gamma$ is a path-ordered 1-cycle, is a topological defect. Furthermore, when $\gamma$ is contractible $T_1(\gamma)$ detects the number of homotopy defects enclosed by $\gamma$. Since Wilson loops of $\widetilde{H}$ gauge theory are in a one-to-one correspondence with the irreducible representations of $\widetilde{H}$, $T_1$ generates a $\mathsf{Rep}(\widetilde{H})$ 1-form symmetry, which we denote as $\mathsf{Rep}(\widetilde{H})^{(1)}$. By 1-gauging $T_1$, we can find topological surface defects that generate 0-form symmetries [128], and the symmetry category describing both the 0-form and 1-form symmetries is $\mathcal{S} = 2\text{-}\mathsf{Rep}(\widetilde{H})$ [27].

Since $\mathcal{S}$ is a finite symmetry, its homotopy defects are its symmetry defects. Therefore, we can diagnose the spontaneous symmetry breaking of $\mathsf{Rep}(\widetilde{H})^{(1)}$ using $T_1$. When the Wilson loop corresponds to a gapped string in the spectrum—the gauge charges are gapped—the $\mathsf{Rep}(\widetilde{H})^{(1)}$ symmetry is spontaneously broken since its homotopy defects are gapped. However, if the Wilson loop is proliferated—the gauge charges are condensed—the $\mathsf{Rep}(\widetilde{H})^{(1)}$ symmetry is not spontaneously broken.

An effective theory that describes the $\mathsf{Rep}(\widetilde{H})^{(1)}$ SSB phase transition is $\widetilde{H}$ gauge theory with an $\widetilde{\mathcal{O}}$ Higgs term and a $\widetilde{G}$ symmetry transforming $\widetilde{\mathcal{O}}$. In the Higgs phase, the gauge charges condense, generating a $G \xrightarrow{\text{ssb}} H$ SSB pattern and ensuring that $\mathcal{S}$ is not spontaneously broken. It, therefore, corresponds to the ordered phase. On the other hand, in the deconfined phase, the gauge charges are gapped, so the $\widetilde{G}$ symmetry is not spontaneously broken but $\mathsf{Rep}(\widetilde{H})^{(1)}$ is. Therefore, the deconfined phase is the defect-suppressed disordered phase.

As an example, let us consider $G = SO(3)$, whose universal covering space is $\widetilde{G} = \mathrm{Spin}(3) \simeq SU(2)$, and arbitrary $H$.[11] $\widetilde{\mathcal{O}}_i$ are $SU(2)$ rotors, but since $SU(2) \cong S^3$, it is convenient to represent them as $\widetilde{\mathcal{O}}_i \in \mathbb{C}^2$ satisfying $\widetilde{\mathcal{O}}_i^\dagger \widetilde{\mathcal{O}}_i = 1$. The gauge field $a_{ij}$ transforms in the fundamental representation of $SU(2)$ restricted to $\widetilde{H}$, and the original $SO(3)$ symmetry is realized as an $SU(2)$ symmetry transforming $\widetilde{\mathcal{O}}_i$. The effective Euclidean action

---

[11] The finite subgroups $H$ of $G = SO(3)$ are the cyclic groups $\mathbb{Z}_n$, the dihedral groups $D_n$ of order $2n$, the tetrahedral group $T$, the octahedral group $O$, and the icosahedral group $I$. Therefore, the possible $\widetilde{H}$ are the cyclic groups $\mathbb{Z}_n$, the binary dihedral groups $2D_n$ of order $4n$, the binary tetrahedral group $2T$, the binary octahedral group $2O$, and the binary icosahedral group $2I$. The particular case of $H = D_2 \simeq \mathbb{Z}_2 \times \mathbb{Z}_2$, where $2D_2 \simeq Q_8$, was discussed in Sec. III C and Ref. 103.

is

$$S = K \sum_{(ijk)} \text{Tr}[(\delta a)_{ijk}] - J \sum_{(ij)} \widetilde{\mathcal{O}}_i^\dagger a_{ij} \widetilde{\mathcal{O}}_j, \qquad (35)$$

where the first term penalizes codimension 2 homotopy defects while the second term is the Higgs term. When $J \gg K$, the model is in the Higgs phase and the gauge charges created by $\widetilde{\mathcal{O}}$ are condensed. Because $\widetilde{\mathcal{O}}$ transforms under $\widetilde{G}$, the Higgs phase spontaneously breaks $\widetilde{G}$. Due to the $\widetilde{H}$ gauge redundancy, the physical symmetry spontaneously broken is $\widetilde{G}/\widetilde{H}$, and the SSB pattern is equivalent to $G \xrightarrow{\text{ssb}} H$. When $J \ll K$, it is in the deconfined phase of $\widetilde{H}$ gauge theory. The gauge charges are gapped and deconfined excitations, so $\widetilde{G}$ is not spontaneously broken but the emergent $\text{Rep}(\widetilde{H})^{(1)}$ symmetry is.

For general $G$, the $\text{Rep}(\widetilde{H})^{(1)}$ SSB phase has non-chiral bosonic topological order, which is the same topological order in the quantum double model $\mathbf{D}(\widetilde{H})$ (see Sec. III C), enriched by the $G$ symmetry. When $\widetilde{H}$ is abelian, it is an abelian topological order, but when $\widetilde{H}$ is non-abelian and $\text{Rep}(\widetilde{H})^{(1)}$ is a non-invertible symmetry, there will be non-abelian anyons. The anyons, which at low energies are the topological defect lines, are described by the braided fusion category $\mathcal{Z}(\text{Vec}_{\widetilde{H}})$. Because there are more topological defect lines at low energies than those described by $\text{Rep}(\widetilde{H})$, there are additional emergent 1-form symmetries. The symmetry category for these topological orders was explored in Ref. 157. These new symmetries arise from the homotopy defects of $\text{Rep}(\widetilde{H})^{(1)} \xrightarrow{\text{ssb}} 1$, just as the $\text{Rep}(\widetilde{H})^{(1)}$ symmetry arose from the homotopy defects of $G \xrightarrow{\text{ssb}} H$. In Eq. (35), this new emergent symmetry is exact in the limit $K \to \infty$ and corresponds to the transformation $a_{ij} \to \widetilde{h}_{ij} a_{ij}$ where $(\delta \widetilde{h})_{ijk} = 1$ and $\widetilde{h}_{ij} \in \widetilde{H}$.

## B. Suppressing hedgehogs

The simplest example of an emergent continuous symmetry that can spontaneously break arises from homotopy defects in $D = 4$ called hedgehogs that occur when $\pi_1(\mathcal{M}) = 0$ while $\pi_2(\mathcal{M}) = \mathbb{Z}$. Using the results from Sec. III A, they are associated with an emergent $U(1)$ 1-form symmetry. An order parameter manifold $\mathcal{M}$ that produces this is the Eilenberg-MacLane space $K(\mathbb{Z}, 2) \simeq \mathbb{CP}^\infty$. However, we will consider the more physically relevant $\mathcal{M} = S^2$ that arises from the SSB pattern $SO(3) \xrightarrow{\text{ssb}} U(1)$ in isotropic ferromagnets and focus only on the codimension 3 homotopy defects.

An $S^2$ order parameter is typically parametrized by $\mathcal{O}_i \in \mathbb{R}^3$ subject to the constraint $|\mathcal{O}_i|^2 = 1$, as in the $O(3)$ sigma model. We construct the $\widetilde{\mathcal{O}}$ presentation of $\mathcal{O}$ using the double covering space $\widetilde{G} = SU(2)$ of $G = SO(3)$

since $\widetilde{G} = SU(2) \cong S^3$ satisfies $\pi_2(\widetilde{G}) = 0$, which lifts $H = U(1)$ to $\widetilde{H} = U(1)$. As in Sec. IV A, we'll represent the $SU(2)$ rotors $\widetilde{\mathcal{O}}_i$ by $\widetilde{\mathcal{O}}_i \in \mathbb{C}^2$ satisfying $\widetilde{\mathcal{O}}_i^\dagger \widetilde{\mathcal{O}}_i = 1$, so the original $SO(3)$ symmetry is realized as an $SU(2)$ symmetry transforming $\widetilde{\mathcal{O}}_i$. Since $\widetilde{H} = U(1)$, there is also a $U(1)$ gauge field $a_{ij} = -a_{ji}$ and gauge redundancy

$$\widetilde{\mathcal{O}}_i \sim e^{i\lambda_i} \widetilde{\mathcal{O}}_i, \qquad a_{ij} \sim a_{ij} + (d\lambda)_{ij}, \qquad (36)$$

where $(d\lambda)_{ij} = \lambda_j - \lambda_i$ and the gauge parameter $\lambda_i \in \mathbb{R}/2\pi\mathbb{Z}$. This enforces physical $\widetilde{\mathcal{O}}$ configurations to correspond to $\mathcal{O}$ configurations. We note that $\widetilde{\mathcal{O}}_i$ and $a_{ij}$ are precisely the degrees of freedom of the $\mathbb{CP}^1$ presentation of the $O(3)$ sigma model [158].

Hedgehogs are strings in $D = 4$ spacetime and reside on the 1-simplices of the dual lattice, which correspond to 3-simplices of the direct lattice. They are detected by $\mathcal{O}$'s homotopy class along a 2-cycle $\Sigma_2$, which in the $\widetilde{\mathcal{O}}$ presentation is encoded by $\frac{1}{2\pi} \sum_{(ijk) \in \Sigma_2} f_{ijk} \in \mathbb{Z}$ where $f_{ijk} = (da)_{ijk} = a_{ij} + a_{jk} - a_{ik}$ is a $U(1)$ 2-cocycle. Therefore, a hedgehog resides on a 3-simplex $(ijkl)$ if $(df)_{ijkl} \neq 0$, so they correspond to 't Hooft lines—magnetic monopole worldlines—of the gauge theory.

While we have managed to represent the homotopy defects by the gauge fields, they manifest themselves through the topology of the gauge group $\widetilde{H} = U(1)$ via $\pi_1(\widetilde{H}) = \mathbb{Z}$. So, we still do not have control over their dynamics. Therefore, just as we presented the $\mathcal{O}_i$ order parameter using $\widetilde{\mathcal{O}}_i$, we must choose a new presentation of the gauge field $a_{ij}$. To trivialize $\pi_1(\widetilde{H})$, we take the universal cover of $\widetilde{H}$ and consider $\mathbb{R}$ gauge fields $\widetilde{a}_{ij}$. To ensure physical $\widetilde{a}_{ij}$ configurations correspond to $a_{ij}$ configurations, we introduce a $\mathbb{Z}$ 2-cochain gauge field $n_{ijk}$ and enhance the gauge redundancy (36) to

$$\begin{aligned} \widetilde{\mathcal{O}}_i &\sim e^{i\widetilde{\lambda}_i} \widetilde{\mathcal{O}}_i, \\ \widetilde{a}_{ij} &\sim \widetilde{a}_{ij} + (d\widetilde{\lambda})_{ij} + 2\pi m_{ij}, \\ n_{ijk} &\sim n_{ijk} + (dm)_{ijk}, \end{aligned} \qquad (37)$$

where the gauge parameters $\widetilde{\lambda}_i \in \mathbb{R}$ and $m_{ij} \in \mathbb{Z}$. In this new presentation, the hedgehogs are the gauge fluxes of $n_{ijk}$ and appear as violations of the flatness condition $(dn)_{ijkl} = 0$.

When the hedgehogs are gapped and cannot end, the ground state always satisfies $(dn)_{ijkl} = 0$ on all 3-simplices. There is then the topological defect surface

$$T_2(\Sigma_2) = e^{i\alpha \sum_{(ijk) \in \Sigma_2} n_{ijk}}. \qquad (38)$$

Since $\sum_{(ijk) \in \Sigma_2} n_{ijk} \in \mathbb{Z}$, which is the number of hedgehogs in $\Sigma_2$, $\alpha \in \mathbb{R}/2\pi\mathbb{Z}$ and $T_2$ generates the aforementioned $U(1)$ 1-form symmetry. In the language of gauge theory, this corresponds to the magnetic symmetry.

Using this presentation of the order parameter, we construct the effective Euclidean action

$$S = K \sum_{(ijkl)} (\mathrm{d}n)^2_{ijkl} - g \sum_{(ijk)} (\widetilde{f}_{ijk} - 2\pi n_{ijk})^2$$
$$- J \sum_{(ij)} \widetilde{\mathcal{O}}_i^\dagger \, \mathrm{e}^{\mathrm{i}\widetilde{a}_{ij}} \widetilde{\mathcal{O}}_j + \text{h.c.}. \tag{39}$$

The first term penalizes Hedgehog configurations,[12] the second is a $U(1)$ Maxwell term, and the third is a Higgs term. Let's assume that $K > 0$ is large and consider the phases of $S$ when tuning $g/J$. The phase diagram resembles model (35) in Sec. IV A. When $J \gg g$, it is in the Higgs phase because $\widetilde{\mathcal{O}}_i$ condenses, but this also gives rise to the SSB pattern $SO(3) \xrightarrow{\text{ssb}} U(1)$ and corresponds to the ordered phase. Indeed, $\widetilde{\mathcal{O}}$ and $\mathcal{O}$ are related by Hopf's map $\mathcal{O}^a = \widetilde{\mathcal{O}}^\dagger \sigma^a \widetilde{\mathcal{O}}$, where $\{\sigma^a\}$ are the Pauli matrices, so if $\widetilde{\mathcal{O}}$ condenses so does $\mathcal{O}$. When $J \ll g$, the Maxwell term dominates and the model is in a Coulomb phase where the gauge charges are gapped, so $SO(3)$ symmetry is not spontaneously broken, but the emergent $U(1)^{(1)}$ symmetry is. This is the defect-suppressed disordered phase.

The $U(1)^{(1)} \xrightarrow{\text{ssb}} 1$ SSB pattern in the defect-suppressed disordered phase generates gapless Goldstone modes that correspond to the photons of the Coulomb phase [7]. Furthermore, it will generate new homotopy defects and, therefore, new emergent symmetries. As discussed in Sec. III B, its homotopy defects are the Wilson loops, and the new emergent symmetry is the $U(1)^{(1)}_e$ electric symmetry. It is not an exact symmetry because Wilson loops can end on gauge charges. So, it emerges in the Coulomb phase below the gauge charges gap. Notice that when $J = 0$, this is an exact symmetry and corresponds to the transformation $a_{ij} \to a_{ij} + \Gamma_{ij}$ where $(\mathrm{d}\Gamma)_{ijk} = 0$.

# V. 'T HOOFT ANOMALIES

Emergent symmetries can have 't Hooft anomalies, which can be self-anomalies or mixed anomalies between emergent and exact symmetries. Formally, this means that the low-energy effective theory with background gauge fields turned on violates gauge invariance by a phase that local counterterms cannot remove. However, this phase can be canceled by an SPT state in one higher dimension through anomaly-inflow, allowing 't Hooft anomalies to be characterized by SPTs. In this section, we investigate the 't Hooft anomalies affiliated with the emergent symmetry $\mathcal{S}$ we have been discussing.

---

[12] The $K \to \infty$ limit is completely hedgehog-free. This can alternatively be achieved using a 1-cochain Lagrange multiplier that sets $(\mathrm{d}n)_{ijkl} = 0$, as is done in generalized Villain models [106].

## A. Mixed anomalies

The fact that $G$ homotopy defects carry $\mathcal{S}$ symmetry charge suggests there is a mixed 't Hooft anomaly between $G$ and $\mathcal{S}$ [159]. Instead of striving for a rigorous understanding with general $\mathcal{S}$, we conjecture that this is indeed always true. The $(D + 1)$-dimensional SPT characterizing this mixed anomaly can be constructed by proliferating "decorated defects" [55, 160–162], particularly $\mathcal{S}$ homotopy defects consistently decorated by $G$ SPTs. We will support our conjecture for general $\mathcal{S}$ using physical reasoning and by considering examples.

A primary motivation for our conjecture is a physical consequence of it. Due to anomaly matching, a mixed 't Hooft anomaly between $G$ and $\mathcal{S}$ prevents the ground state from being trivial whenever both $G$ and $\mathcal{S}$ exist. This obstruction exists whenever $\mathcal{S}$ can emerge. Therefore, the ground state cannot be trivial whenever there are gapped homotopy defects. This is, in fact, consistent with the widely believed folklore that to transition from an ordered phase to the trivial disordered phase, one must proliferate all flavors of homotopy defects [163–168]. Indeed, proliferating all of the homotopy defects would prevent $\mathcal{S}$ from ever emerging, destroying any 't Hooft anomalies involving $\mathcal{S}$ and removing the obstruction.

The nature of the defect-suppressed disordered phase transition discussed in Sec. IV provides additional evidence supporting this conjecture. It is a direct transition between two distinct SSB patterns, one where $G \xrightarrow{\text{ssb}} H$ and $\mathcal{S}$ is unbroken and another where $G$ is unbroken but $\mathcal{S}$ is (see Fig. 3). The transition from the defect-suppressed disordered phase to the ordered phase is driven by proliferating the $\mathcal{S}$ homotopy defects. However, proliferating these homotopy defects must also condense $G$ symmetry charges to ensure $G \xrightarrow{\text{ssb}} H$, so the $\mathcal{S}$ homotopy defects must be decorated by $G$ symmetry charges. When $\mathcal{S}$ is an invertible 0-form symmetry and the transition is an ordinary DQCP, the decoration is a manifestation of a mixed anomaly between the symmetries [169–173]. It is natural to expect that when $\mathcal{S}$ is a generalized symmetry, this decoration remains a manifestation of a mixed anomaly between $\mathcal{S}$ and $G$.

Let us now consider a simple scenario where the microscopic symmetry group $G$ is a Lie group and $\mathcal{S}$ describes a finite $(D-2)$-form symmetry. The codimension 2 $G$ homotopy defects charged under $\mathcal{S}$ are trivialized in the $\widetilde{\mathcal{O}}$ presentation by taking $\widetilde{G}$ to be the universal cover of $G$. When $\widetilde{\mathcal{O}}$ transforms in a finite-dimensional irreducible unitary representation of $\widetilde{G}$, which corresponds to an irreducible projective representation of $G$ [174], it carries fractional $G$ symmetry charge [144, 175, 176]. This also can apply for a continuous $(D-3)$-form symmetry, as we saw in Sec. IV B. In both of these cases, the $\mathcal{S}$ homotopy defects—the defects that proliferate to restore $\mathcal{S}$—are Wilson lines in the $\widetilde{\mathcal{O}}$ presentation. Since the Wilson lines can end on $\widetilde{\mathcal{O}}$, they too carry fractional $G$ symmetry charge, which is a manifestation of a mixed 't

Hooft anomaly between $G$ and $\mathcal{S}$ [177–180].

The examples in Sec IV with $G = SO(3)$ all had mixed anomalies that manifested themselves as symmetry fractionalization. Indeed, since $\widetilde{\mathcal{O}}$ transformed under the fundamental representation of $\widetilde{G} = SU(2)$, The Wilson lines carried spin $1/2$—fractional $SO(3)$ symmetry charge.

Another general scenario is when $\mathcal{S}$ is a continuous symmetry such that it has a conserved Noether current. A sufficient, but not necessary, condition to diagnose a mixed anomaly between $G$ and $\mathcal{S}$ is through a particular violation of the current conservation law in the presence of $G$ background fields. For $U(1)$ higher-form symmetries, this Noether current will be the topological currents $J^{\text{top}}$ discussed in Sec. III A. In this case, it was shown by Ref. 92 that a general class of mixed anomalies between $G$ and the $U(1)$ higher-form symmetries exists that manifest in violations of $\mathrm{d} * J^{\text{top}} = 0$.

We now consider some simple examples of the emergent mixed 't Hooft anomaly in ordered phases for $D = 4$. We'll restrict to examples where $\mathcal{S}$ is an invertible symmetry so the SPT that characterizes the mixed anomaly can be easily discussed.

### 1. $Z_N$ ferromagnet

Let us first consider a $Z_N$ ferromagnet, an ordered phase where $\mathbb{Z}_N \xrightarrow{\text{ssb}} 1$. There are codimension 1 homotopy defects classified by $\mathbb{Z}_N$ and, in $D = 4$, an emergent $\mathbb{Z}_N^{(3)}$ symmetry in the ground state subspace. Since $\mathbb{Z}_N^{(3)}$ cannot spontaneously break in $D = 4$, there is no defect-suppressed disordered phase.

The ground state subspace is described by the topological field theory

$$S = \frac{2\pi \mathrm{i}}{N} \int_{\mathbb{M}_4} b \cup \mathrm{d}\phi, \qquad (40)$$

where $\mathbb{M}_4$ is a triangulation of spacetime, $\phi$ is a $\mathbb{Z}_N$-valued 0-cochain, $\cup$ is the cup product, d is the simplicial codifferential, and $b$ is a $\mathbb{Z}_N$-valued 3-cochain. The microscopic $\mathbb{Z}_N$ symmetry is generated by the topological defect

$$T_3(\Sigma_3) = \mathrm{e}^{\frac{2\pi \mathrm{i}}{N} \sum_{(ijkl) \in \Sigma_3} b_{ijkl}}, \qquad (41)$$

where $\Sigma_3$ is a 3-cycle, and the emergent $\mathbb{Z}_N^{(3)}$ symmetry is generated by the local topological defect

$$T_0(i) = \mathrm{e}^{\frac{2\pi \mathrm{i}}{N} \phi_i}. \qquad (42)$$

$T_0$ supported on a 0-sphere $T_0(S^0) = \mathrm{e}^{\frac{2\pi \mathrm{i}}{N} \sum_{i \in S^0} \phi_i}$ detects the codimension 1 homotopy defects. It equals one in their absence, which signals that the $\mathbb{Z}_N$ symmetry is spontaneously broken.

It is well known that the $\mathbb{Z}_N$ and $\mathbb{Z}_N^{(3)}$ symmetries in (40) realize a mixed 't Hooft anomaly [62, 66, 88]. A

manifestation of this anomaly is seen in the correlation function

$$\langle T_3(\Sigma_3) T_0(S^0) \rangle = \mathrm{e}^{\frac{2\pi \mathrm{i}}{N} \text{link}(\Sigma_3, S^0)}. \qquad (43)$$

Going to five-dimensional spacetime $\mathbb{M}_5$ and turning on background $\mathbb{Z}_N$ and $\mathbb{Z}_N^{(3)}$ gauge fields $A^{(1)} \in H^1(\mathbb{M}_5, \mathbb{Z}_N)$ and $B^{(4)} \in H^4(\mathbb{M}_5; \mathbb{Z}_N)$, the anomaly is characterized by the SPT

$$S_{\text{SPT}} = \frac{2\pi \mathrm{i}}{N} \int_{\mathbb{M}_5} A^{(1)} \cup B^{(4)}. \qquad (44)$$

It is straightforward to generalize this to the SSB pattern $\mathbb{Z}_N^{(p)} \xrightarrow{\text{ssb}} 1$ in $D$-dimensional spacetime. The emergent symmetry is now $\mathbb{Z}_N^{(D-p-1)}$ and the low-energy effective theory has the same form as Eq. (40), but with $\phi$ and $b$ replaced by $\mathbb{Z}_N$-valued $p$ and $(D - p - 1)$-cochains, respectively. There is still a mixed 't Hooft anomaly, which is characterized by a topological action like Eq. (44) but with $A$ and $B$ now $\mathbb{Z}_N$ $(p + 1)$ and $(D - p)$-cocycles, respectively.

### 2. Superfluid

A bosonic superfluid is an ordered phase where $U(1) \xrightarrow{\text{ssb}} 1$. The order parameter manifold is $\mathcal{M} = S^1$, so in $D = 4$ there codimension 2 homotopy defects classified by $\mathbb{Z}$ and an emergent $U(1)^{(2)}$ symmetry. Since $U(1)^{(2)}$ cannot spontaneously break in $D = 4$, there is no defect-suppressed disordered phase.

The ordered phase at long wavelengths is described by the $S^1$ nonlinear sigma model (NLSM)

$$S = \frac{1}{2g^2} \int_{M_4} \mathrm{d}\mathcal{O} \wedge * \mathrm{d}\mathcal{O}, \qquad (45)$$

where $\mathcal{O} : M_4 \to \mathbb{R}/2\pi\mathbb{Z}$. The Noether current of the microscopic $U(1)$ symmetry is $*j = \frac{1}{g^2} * \mathrm{d}\mathcal{O}$, so the $U(1)$ symmetry is generated by the topological defect

$$T_3^{(\alpha)}(M) = \mathrm{e}^{\frac{\mathrm{i}\alpha}{g^2} \int_M * \mathrm{d}\mathcal{O}}. \qquad (46)$$

The topological defect line that generates the emergent $U(1)^{(2)}$ symmetry is

$$T_1^{(\beta)}(C) = \mathrm{e}^{\frac{\mathrm{i}\beta}{2\pi} \int_C \mathrm{d}\mathcal{O}}, \qquad (47)$$

which detects codimension 2 vortices linking with $C$.

It is well known that the $U(1)$ and $U(1)^{(2)}$ symmetries form a mixed anomaly [63, 67, 69, 88, 92, 181]. A manifestation of it is in the correlation function

$$\langle T_3^{(\alpha)}(M) T_1^{(\beta)}(C) \rangle = \mathrm{e}^{\frac{\mathrm{i}\alpha\beta}{2\pi} \text{link}(M, \partial C)}. \qquad (48)$$

Going to 5-dimensional spacetime $M_5$ and turning on $U(1)$ and $U(1)^{(2)}$ background gauge fields $A^{(1)}$ and $B^{(3)}$, the anomaly is characterized by the SPT

$$S_{\text{SPT}} = \mathrm{i} \int_{M_5} B^{(3)} \wedge \frac{\mathrm{d}A^{(1)}}{2\pi}. \qquad (49)$$

This action reveals that the anomaly can be detected by turning on $A^{(1)}$ and observing that the $U(1)^{(2)}$ symmetry's Noether current is no longer conserved and violated by $* \frac{\mathrm{d}A^{(1)}}{2\pi}$.

It is again straightforward to generalize this to $U(1)^{(p)} \xrightarrow{\text{ssb}} 1$ in $D$-dimensions. $\mathcal{O}$ is replaced with a $p$-form $U(1)$ gauge field, causing the $S^1$ NLSM (45) to become $p$-form Maxwell theory. The SPT takes the same form as Eq. (49), but with $A$ and $B$ now $(p+1)$-form and $(D - p - 1)$-form fields, respectively.

### 3. Isotropic ferromagnet

An isotropic ferromagnet in $D = 4$ is an ordered phase with the SSB pattern $SO(3) \xrightarrow{\text{ssb}} U(1)$, which we considered in Sec. IV B. Its order parameter manifold is $\mathcal{M} = S^2$, so its homotopy defects are hedgehogs and there is an emergent $U(1)^{(1)}$ symmetry. At long wavelength, it is described by the $S^2$ NLSM

$$S = \frac{1}{2g^2} \int_{M_4} \mathrm{d}\mathcal{O} \wedge * \, \mathrm{d}\mathcal{O}, \tag{50}$$

where $\mathcal{O} : M_4 \to S^2$. It is convenient to parametrize $S^2$ using the unit vector $\boldsymbol{n} \in \mathbb{R}^3$ and consider the $O(3)$ sigma model

$$S = \frac{1}{2g^2} \int_{M_4} \mathrm{d}^4 x \; |\boldsymbol{\nabla} \boldsymbol{n}|^2. \tag{51}$$

The topological defect surface generating the $U(1)^{(1)}$ symmetry is

$$T_2^{(\alpha)}(\Sigma_2) = \mathrm{e}^{\frac{\mathrm{i}\alpha}{8\pi} \int_{\Sigma_2} \mathrm{d}S_{ij} \; \epsilon^{ijkl} \boldsymbol{n} \cdot (\partial_k \boldsymbol{n} \times \partial_l \boldsymbol{n})}. \tag{52}$$

As mentioned towards the beginning of this section, the $U(1)^{(1)}$ symmetry's homotopy defects carry fractional $SO(3)$ symmetry charge, so there is a mixed anomaly between the $SO(3)$ and $U(1)^{(1)}$ symmetries. Going to a triangulated 5-dimensional spacetime $\mathbb{M}_5$ and turning on background $SO(3)$ and $\mathbb{Z}_2^{(1)} \subset U(1)^{(1)}$ gauge fields $A^{(1)}$ and $B^{(2)} \in H^2(\mathbb{M}_5; \mathbb{Z}_2)$, respectively, this symmetry fractionalization anomaly is charactered by the SPT [178]

$$S_{\mathrm{SPT}} = \mathrm{i}\pi \int_{\mathbb{M}_5} \omega_2(A^{(1)}) \cup \beta(B^{(2)}), \tag{53}$$

where $\omega_2(A^{(1)}) \in H^2(\mathbb{M}_5, \mathbb{Z}_2)$ is the 2nd Stiefel-Whitney class of the $SO(3)$ bundle and $\beta(B^{(2)}) \in H^3(\mathbb{M}_5, \mathbb{Z}_2)$ is the image of Bockstein homomorphism

$$\beta : H^2(\mathbb{M}_5, \mathbb{Z}_2) \to H^3(\mathbb{M}_5, \mathbb{Z}_2). \tag{54}$$

Introducing the lift $\widetilde{B}^{(2)}$ of $B^{(2)}$ to $\mathbb{Z}_4$ coefficients that satisfies $B^{(2)} = \widetilde{B}^{(2)} \bmod 2$, it has the explicit form $\beta(B^{(2)}) = \frac{1}{2} \mathrm{d}\widetilde{B}^{(2)}$. A "decorated defect" construction for this SPT was discussed in Refs. 54 and 55. Furthermore,

given the form of $S_{\mathrm{SPT}}$, this mixed anomaly would not manifest by violating a Noether's current conservation law.

Generalizing to arbitrary $D$-dimensional spacetime is straightforward. The only change is that the emergent symmetry would be $U(1)^{(D-3)}$, so the background field $B$ of the $\mathbb{Z}_2^{(D-3)}$ subgroup will be a representative of $H^{D-2}(\mathbb{M}_{D+1}, \mathbb{Z}_2)$. Consequently, in $S_{\mathrm{SPT}}$, the Bockstein homomorphism would be modified to $\beta : H^{D-2}(\mathbb{M}_{D+1}, \mathbb{Z}_2) \to H^{D-1}(\mathbb{M}_{D+1}, \mathbb{Z}_2)$.

## VI. DISCUSSION

In this paper, we explored the rich landscape of generalized symmetries that emerge in generic ordered phases. This provides a general physical setting relevant to numerous fields in physics in which generalized symmetries appear and have physical consequences. We explored the categorical description of these symmetries in Sec. III and discussed their spontaneous symmetry breaking in Sec. IV and their 't Hooft anomalies in Sec. V. Here we summarize some interesting future directions.

While we investigated the structure of the emergent symmetry $\mathcal{S}$, we did not generally consider the mathematical structure formed by the microscopic symmetry group $G$ and $\mathcal{S}$. The only instance in which this was considered was in Sec. III D where we found that when $\mathcal{S}$ was invertible and transformed codimension one homotopy defects, $G$ and $\mathcal{S}$ formed a split higher-group. It is natural to wonder if, for general invertible $\mathcal{S}$, the emergent symmetry can mix with $G$ to form a nontrivial higher group. This was studied in Ref. 92 for continuous invertible $\mathcal{S}$, and it would be interesting to investigate finite invertible $\mathcal{S}$ as well. Furthermore, it would also be interesting to consider the general case where $\mathcal{S}$ can be non-invertible.

Throughout the paper, we assumed that the microscopic symmetry $G$ was an internal symmetry. However, it would be interesting to consider spacetime symmetries. In a crystal, where continuous spatial symmetries are spontaneously broken to a discrete subgroup, the homotopy defects are known to have restricted mobility [6] (i.e., they are subdimensional particles [182, 183]). For example, in two-dimensional space, dislocations and disclinations are lineons and fractons, respectively, due to a dipole moment conservation law [184]. Therefore, the emergent symmetry $\mathcal{S}$ would be a dipole symmetry [185, 186]. It would be interesting to see if there are crystals, or other ordered phases, with emergent multipole symmetries [187–189] or subsystem symmetries.

Lastly, as noted in Sec. IV, the transition between the ordered and defect-suppressed disordered phases may be a deconfined quantum critical point (DQCP) involving generalized symmetries. Studying this phase transition and DQCPs that spontaneously break only generalized symmetries in greater detail would be interesting. For example, whereas DQCPs of ordinary symmetries

can be generally understood using Wess–Zumino–Witten terms [173, 190, 191], it would be interesting to understand possible analogs for generalized symmetries.

*Note added.* Upon completion of this work, we noticed a recent independent work [192] in which the mathematical structure of the generalized symmetries associated with homotopy defects was also studied. Where our results overlap, they agree.

## ACKNOWLEDGEMENTS

We thank Ho Tat Lam and Xiao-Gang Wen for their comments on the manuscript and Lea Bottini, Arkya Chatterjee, Thomas Dumitrescu, Zhi-Qiang Gao, Hart Goldman, Yu Leon Liu, Da-Chuan Lu, Sakura Schäfer-Nameki, Nathan Seiberg, Ryan Thorngren, Senthil Todadri, Xiao-Gang Wen, and Carolyn Zhang for related discussions. This work is supported by the National Science Foundation Graduate Research Fellowship under Grant No. 2141064 and the Henry W. Kendall Fellowship.

## Appendix A: Based and free homotopy classes

In Sec. II A, we discussed the classification of homotopy defects in SSB phases. While homotopy defects are classified by $[\Sigma_k, \mathcal{M}]_\mathrm{f}$, the *free* homotopy classes of based maps $\Sigma_k \to \mathcal{M}$, it is convenient to consider the *based* homotopy classes $[\Sigma_k, \mathcal{M}]_\mathrm{b}$. Here we will briefly compare these two types of homotopy. We refer the reader to Sec. 1.4 of Ref. 193 for a more rigorous discussion.

A map $f : A \to B$ is a based map if $A$ and $B$ are based spaces and $f$ maps the basepoint of $A$ to the basepoint of $B$. Therefore, if the basepoints of $A$ and $B$ are $a$ and $b$, respectively, then $f(a) = b$. Using the based map $f$, we obtain a homotopy $h : A \times I \to B$, and letting $t$ parametrize $I$, since $f$ is a based map $h(a, t = 0) = h(a, t = 1) = b$. However, $h$ need not map $a$ to $b$ for $0 < t < 1$. If it does, then $h$ is a *based* homotopy, and if it doesn't, then it is a *free* homotopy.

Two maps $f_1 : A \to B$ and $f_2 : A \to B$ that are freely homotopic may not be based homotopic. Put differently, letting $[f_i]_\mathrm{f} \in [A, B]_\mathrm{f}$ and $[f_i]_\mathrm{b} \in [A, B]_\mathrm{b}$, it is possible that $[f_1]_\mathrm{f} = [f_2]_\mathrm{f}$ while $[f_1]_\mathrm{b} \neq [f_2]_\mathrm{b}$. Therefore, the number of based homotopy classes is always greater than or equal to the number of free homotopy classes. A simple example of this is the based map $S^1 \to S^1 \vee S^1$, as shown in Fig. 5.

In the context of homotopy defects detected by $\Sigma_k \simeq S^k$, letting $A = S^k$ and $B = \mathcal{M}$, this means that the set of free homotopy classes $[S^k, \mathcal{M}]_\mathrm{f}$ is a subset of the based homotopy classes $\pi_k(\mathcal{M})$. So, the $k$th homotopy group of $\mathcal{M}$ does not directly classify codimension $k + 1$ homotopy defects [95].

Free homotopy classes of $f : A \to B$ can be found from the based homotopy classes of $f$. Indeed, under a free

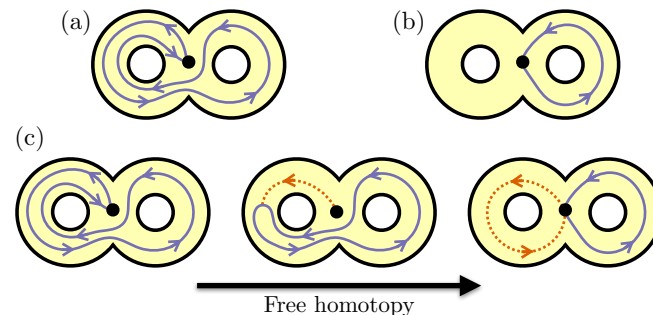

FIG. 5. The based maps $\gamma : S^1 \to S^1 \vee S^1$ in (a) and (b) are not based homotopic, (c) but are freely homotopic. Under the free homotopy, the base point of $S^1$ forms a loop based at the base point of $S^1 \vee S^1$.

homotopy $h : A \times I \to B$ between two based maps $f_1$ and $f_2$ that are not based homotopic, the base point $a \in A$ forms a loop based at the base point $b \in B$ (e.g., Fig. 5c). In terms of the free homotopy $h$, it implies that $h(a, t) = \alpha(t)$, and since $h$ is a homotopy, $\alpha$ depends only on its class $[\alpha]_\mathrm{b} \in \pi_1(B, b)$. This defines an action of $\pi_1(B)$ on $[A, B]_\mathrm{b}$ that connects freely homotopic elements of $[A, B]_\mathrm{b}$, and provides a one to one correspondence

$$[A, B]_\mathrm{b}/\pi_1(B) \leftrightarrow [A, B]_\mathrm{f}. \tag{A1}$$

Setting $A = S^k$ and $B = \mathcal{M}$ yields Eq. (7) in the main text.

The action of $\pi_1(B)$ on $[A, B]_\mathrm{b}$ is defined by Eq. (A1), but it does not have a general closed expression. However, when $A = S^1$, the action of $\pi_1(B)$ is easy to understand. Consider based loops $\gamma_1 : S^1 \to B$ and $\gamma_2 : S^1 \to B$ such that $[\gamma_1]_\mathrm{f} = [\gamma_2]_\mathrm{f}$ but $[\gamma_1]_\mathrm{b} \neq [\gamma_2]_\mathrm{b}$. Furthermore, suppose that under the free homotopy between $\gamma_1$ and $\gamma_2$, the $S^1$ base point's path forms the loop $\alpha \in \pi_1(B)$. For instance, $\gamma_1$ can be the loop in Fig. 5a, $\gamma_2$ in Fig. 5b, and $\alpha$ in Fig. 5c. While $\gamma_1$ and $\gamma_2$ are not be based homotopic, $\gamma_1$ and $\alpha^{-1} \circ \gamma_2 \circ \alpha$ generally are. Therefore, $[\gamma_1]_\mathrm{b} = [\alpha^{-1} \circ \gamma_2 \circ \alpha]_\mathrm{b}$, which letting $\cdot$ denote the product in the fundamental group $\pi_1(B)$, $[\gamma_1]_\mathrm{b} = [\alpha^{-1}]_\mathrm{b} \cdot [\gamma_2]_\mathrm{b} \cdot [\alpha^{-1}]_\mathrm{b}^{-1}$. So, the action of $\pi_1(B)$ on $[S^1, B]_\mathrm{b}$ is given by conjugation. This of course means that the free homotopy classes of $S^1 \to B$ are the same as the conjugacy classes of $\pi_1(B)$

## Appendix B: The symmetry category

As discussed in Sec. I, the modern perspective of symmetries in a many-body system is that they correspond to topological defects [7] (see footnote 1). For invertible 0-form symmetries, the mathematical structure that describes their topological defects fusion is a group. A natural question now arises: what is the mathematical structure that encodes the topological defects' fusion rules Eq. (2) and takes the role of the symmetry group $G$?

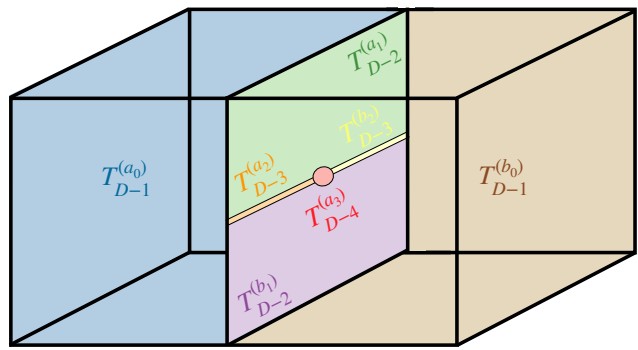

FIG. 6. Shows the layering structure formed by codimension 1, 2, 3, and 4 topological defects.

The expectation is that these properties in $D$-dimensional spacetime can be captured by a $(D-1)$-category $\mathcal{S}$, which we call the symmetry category since it replaces the symmetry group. Using a $(D-1)$-category to describe generalized symmetries is very natural because topological defects of different dimensions form a layering structure, as shown in Fig. 6. Topological defects of different dimension are encoded in $\mathcal{S}$ as follows:

1. The objects of $\mathcal{S}$ are codimension 1 topological defects $\{T_{D-1}^{(a_0)}\}$ and generate the transformations of 0-form symmetries.

2. The 1-morphisms $\mathrm{Hom}(T_{D-1}^{(a_0)}, T_{D-1}^{(b_0)})$ are topological interfaces between $T_{D-1}^{(a_0)}$ and $T_{D-1}^{(b_0)}$. Therefore, they are codimension 2 topological defects $\{T_{D-2}^{(a_1)}\}$ and $\mathrm{Hom}(T_{D-1}^{(\mathbf{1})}, T_{D-1}^{(\mathbf{1})})$ generate 1-form symmetry transformations.

3. The 2-morphisms are topological interfaces between two codimension 2 topological defects $T_{D-2}^{(a,b;A)}$ and $T_{D-2}^{(a,b;B)}$ in $\mathrm{Hom}(T_{D-1}^{(a)}, T_{D-1}^{(b)})$. Therefore, they are codimension 3 topological defects $\{T_{D-3}^{(a_2)}\}$ and $\mathrm{Hom}(T_{D-2}^{(\mathbf{1,1;1})}, T_{D-2}^{(\mathbf{1,1;1})})$ generate 2-form symmetry transformations.

4. One can continue this iteratively for $n$-morphisms $(n \leq D-1)$, which are codimension $n+1$ topological defects and generate to $n$-form symmetries.

Including the fusion property Eq. (2) makes $\mathcal{S}$ into a monoidal $(D-1)$-category. In fact, for general discrete symmetries, $\mathcal{S}$ is a multi-tensor $(D-1)$-category [22, 23, 26, 27, 31]. If there are only finite symmetries, $\mathcal{S}$ reduces to a multi-fusion $(D-1)$-category. Furthermore, if there are no $(D-1)$-form symmetries, $\mathcal{S}$ reduces to a tensor $(D-1)$-category, or if there are only finite symmetries, then a fusion $(D-1)$-category.

The symmetry category of an anomaly-free invertible 0-form symmetry described by the finite group $G$ is $\mathcal{S} = (D-1)\text{-}\mathsf{Vec}_G$. This is the $(D-1)$-category formed by $G$-graded $(D-1)$-vector spaces and includes two types of topological defects:

1. Codimension 1 invertible topological defects $\{T_{D-1}^{(g)}\}$, where $g \in G$, that generate the $G$ 0-form symmetry. These are the objects of $(D-1)\text{-}\mathsf{Vec}_G$.

2. Trivial topological defects of all codimensions larger than 1, which correspond to the $n$-morphisms $(n > 0)$ of $(D-1)\text{-}\mathsf{Vec}_G$.

For a 0-form $G$ symmetry with a 't Hooft anomaly classified by $[\omega] \in H^{D+1}(BG, U(1))$, the symmetry category is $\mathcal{S} = (D-1)\text{-}\mathsf{Vec}_G^{\omega}$, which is the $(D-1)$-category formed by $G$-graded $(D-1)$-vector spaces twisted by the $(D+1)$-cocycle $\omega$.

The most general invertible symmetry is described by a $D$-group $\mathbb{G}^{(D)}$, which contains $r$-form $G^{(r)}$ symmetries of all degrees $0 \leq r \leq D-1$. When all $G^{(r)}$ are finite, and the symmetry is anomaly-free, the symmetry category is $(D-1)\text{-}\mathsf{Vec}_{\mathbb{G}^{(D)}}$ and includes two types of topological defects:

1. Codimension $(r+1)$ invertible topological defects $\{T_{D-r-1}^{(g_r)}\}$, where $g_r \in G^{(r)}$, that generate the $G^{(r)}$ $r$-form symmetries.

2. Topological defects, of various codimension obtained by higher-gauging [128] the $G^{(r)}$ symmetries. These topological defects are called condensation defects and can be invertible or non-invertible.

If there is a 't Hooft anomaly, the symmetry category is instead given by $\mathcal{S} = (D-1)\text{-}\mathsf{Vec}_{\mathbb{G}^{(D)}}^{\omega}$ where $[\omega] \in H^{D+1}(B\mathbb{G}^{(D)}, U(1))$.

Another commonly encountered symmetry category is $\mathcal{S} = (D-1)\text{-}\mathsf{Rep}(G)$, which is a $(D-1)$-category formed by the $(D-1)$-representations of a finite group $G$. It arises, for instance, by gauging a $(D-1)\text{-}\mathsf{Vec}_G$ to produce a new theory with a $(D-1)\text{-}\mathsf{Rep}(G)$ symmetry [18, 22, 27]. As a symmetry category, $(D-1)\text{-}\mathsf{Rep}(G)$ includes two types of topological defects:

1. One dimensional topological defects $\{T_1^{(\pi)}\}$, where $\pi \in \mathsf{Rep}(G)$, which generates a $\mathsf{Rep}(G)$ $(D-2)$-form symmetry. When $G$ is abelian, $\{T_1^{(\pi)}\}$ are invertible and described by the Pontryagin dual of $G$. When $G$ is non-abelian, they are non-invertible.

2. Condensation defects obtained by higher-gauging $\{T_1^{(\pi)}\}$.

More generally, if one gauges a finite $D$-group $\mathbb{G}^{(D)}$ symmetry, the new theory has a symmetry described by the $(D-1)$-representation of the $D$-group, which is captured by $\mathcal{S} = (D-1)\text{-}\mathsf{Rep}(\mathbb{G}^{(D)})$.

## Appendix C: The SymTFT

It is fruitful to describe a symmetry in a way that separates its symmetry category $\mathcal{S}$ from the $D$-dimensional physical theory enjoying the symmetry. For invertible

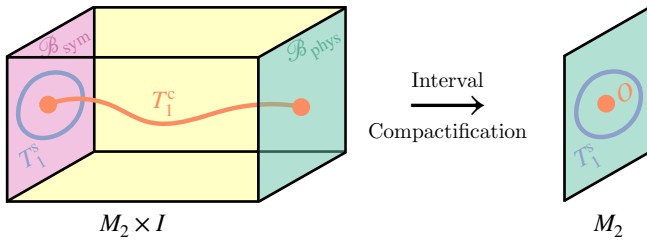

FIG. 7. Upon interval compactification, (left) the symTFT $\mathfrak{Z}(\mathcal{S})$ becomes (right) the physical theory with $\mathcal{S}$ symmetry. The topological defect $T_1^s \in \mathcal{Z}(\mathcal{S})$ is a topological defect on $\mathscr{B}^{\mathrm{sym}}$, so $T_1^s \in \mathcal{S}$. The topological defect $T_1^c \in \mathcal{Z}(\mathcal{S})$ ends on $\mathscr{B}^{\mathrm{sym}}$, so $T_1^c \notin \mathcal{S}$ and instead becomes $O$ after interval compactification and carries $\mathcal{S}$ symmetry charge.

symmetries, this can be accomplished using the representation theory of groups. What about generalized symmetries? When the symmetry is finite (i.e., $\mathcal{S}$ is a multifusion $(D-1)$-category), a useful way to do so is using the symmetry topological field theory (SymTFT) $\mathfrak{Z}(\mathcal{S})$ associated with the symmetry $\mathcal{S}$ [22, 23, 71, 108–126].

$\mathfrak{Z}(\mathcal{S})$ is a TFT in $(D+1)$-dimensional spacetime $M_D \times I$ with special boundary conditions on $\partial I$. There is the topological boundary $\mathscr{B}^{\mathrm{sym}}$ that provides a physical realization of the data in $\mathcal{S}$, independent of the physical theory, and the boundary $\mathscr{B}^{\mathrm{phys}}$ which is not necessarily topological and depends on the details of the theory. $\mathfrak{Z}(\mathcal{S})$ is *defined* by the requirement that it has a topological boundary $\mathscr{B}^{\mathrm{sym}}$ whose topological defects form the symmetry category $\mathcal{S}$. It is related to the physical theory by an interval compactification, as shown in Fig. 7.

Since $\mathfrak{Z}(\mathcal{S})$ has a topological boundary condition described by $\mathcal{S}$, its bulk topological defects are described by the Drinfeld center $\mathcal{Z}(\mathcal{S})$ of $\mathcal{S}$ [117, 118]. The topological defects in $\mathfrak{Z}(\mathcal{S})$ correspond to the topological defects *and* charged objects of $\mathcal{S}$. Only upon imposing the correct topological boundary condition $\mathscr{B}^{\mathrm{sym}}$ can one distinguish which is which. The topological defects on the $\mathscr{B}^{\mathrm{sym}}$ boundary of $\mathfrak{Z}(\mathcal{S})$ (e.g., $T_1^s$ in Fig. 7) are described by $\mathcal{S}$ and become the topological defects of the physical theory after interval compactification. The topological defects that can end on the $\mathscr{B}^{\mathrm{sym}}$ and $\mathscr{B}^{\mathrm{phys}}$ boundaries (e.g., $T_1^c$ in Fig. 7) become the operators charged under $\mathcal{S}$ in the physical theory.

By construction, for two physically distinct theories both with an $\mathcal{S}$ symmetry, their symmetries are both described by the same SymTFT $\mathfrak{Z}(\mathcal{S})$ with the same $\mathscr{B}^{\mathrm{sym}}$ boundary but different $\mathscr{B}^{\mathrm{phys}}$ boundaries. That said, two different symmetries $\mathcal{S}_1$ and $\mathcal{S}_2$ can also have the same SymTFT $\mathfrak{Z}(\mathcal{S})$ if $\mathcal{Z}(\mathcal{S}_1) = \mathcal{Z}(\mathcal{S}_2)$. While their bulks are the same, their topological boundaries $\mathscr{B}^{\mathrm{sym}}_{\mathcal{S}_1}$ and $\mathscr{B}^{\mathrm{sym}}_{\mathcal{S}_2}$ of $\mathfrak{Z}(\mathcal{S})$ would be different. In fact, if two symmetries have the same SymTFT, they are related under gauging [22, 23], and therefore, changing the topological boundary conditions corresponds to gauging the symmetry.

There is a general way to construct $\mathfrak{Z}(\mathcal{S})$ when $\mathcal{S}$

describes an invertible symmetry. As discussed in appendix B, the symmetry category of a general invertible finite symmetry is $\mathcal{S} = (D-1)\text{-}\mathsf{Vec}^{\omega}_{\mathbb{G}^{(D)}}$ where $[\omega] \in H^{D+1}(B\mathbb{G}^{(D)}, \mathbb{R}/\mathbb{Z})$. To construct $\mathfrak{Z}(\mathcal{S})$, we first consider the $(D+1)$-dimensional $\mathcal{S}$-SPT $\mathfrak{T}_{\mathbb{G}^{(D)} \text{ SPT}}$ described by

$$Z_{\mathbb{G}^{(D)} \text{ SPT}} = e^{2\pi \mathrm{i} \int_{M_{D+1}} \mathcal{A}^* \omega}, \qquad (\text{C1})$$

where $M_{D+1}$ is a closed spacetime and the background field $\mathcal{A}$ belongs to a homotopy class $[M_{D+1}, B\mathbb{G}^{(D)}]$. Of course, if $\mathcal{S}$ is anomaly-free (i.e., $[\omega] = [0]$), this is the trivial $\mathbb{G}^{(D)}$ SPT. The SymTFT is found by gauging the SPT $Z_{\mathbb{G}^{(D)} \text{ SPT}}$ (making $\mathcal{A}$ dynamical):

$$\mathfrak{Z}(\mathcal{S}) = \mathfrak{T}_{\mathbb{G}^{(D)} \text{ SPT}}/\mathbb{G}^{(D)}. \qquad (\text{C2})$$

The topological boundary $\mathscr{B}^{\mathrm{sym}}$ for $\mathcal{S}$ will generally be the Dirichlet boundary condition for the $\mathbb{G}^{(D)}$ gauge fields.

When $\mathbb{G}^{(D)}$ is a direct product of abelian finite $r$-form symmetries

$$\mathbb{G}^{(D)} = \prod_{r=0}^{D-1} G^{(r)}, \qquad (\text{C3})$$

with $G^{(r)} = \mathbb{Z}_{N_r}$, we can find a Lagrangian description of Eq. (C2). In this simple case, the $\mathbb{G}^{(D)}$ gauge fields are a collection of $G^{(r)}$ $(r+1)$-cocycles. The SymTFT is the generalized Dijkgraaf-Witten theory

$$Z_{\mathrm{symTFT}} = \sum_{\{a_{r+1}\}} e^{2\pi \mathrm{i} \int_{M_{D+1}} \omega(a_1, \cdots, a_D)}, \qquad (\text{C4})$$

where $a_n \in H^n(M_{D+1}, G^{(n-1)})$. It is convenient to rewrite this as

$$Z_{\mathrm{symTFT}} = \sum_{\{c_{r+1}, b_{D-r-1}\}} e^{2\pi \mathrm{i} \int_{M_{D+1}} \sum_r \frac{c_{r+1} \cup \mathrm{d}b_{D-r-1}}{N_r} + \omega(c_1, \cdots, c_D)}, \qquad (\text{C5})$$

where the sum is over $c_n \in C^n(M_{D+1}, G^{(n-1)})$ and $b_n \in C^n(M_{D+1}, \hat{G}^{(D-n-1)})$, with $\hat{G}^{(r)}$ is the Pontryagin dual of $G^{(r)}$, and $\mathrm{d}$ and $\cup$ are the simplicial codifferential and cup product, respectively. The equations of motion are

$$\frac{2\pi}{N_r} \mathrm{d}c_{r+1} = 0, \qquad (\text{C6})$$

$$\mathrm{d}b_{D-r-1} = N_r \frac{\delta}{\delta c_{r+1}} \omega(c_1, \cdots, c_{r+1}, \cdots, c_D). \qquad (\text{C7})$$

We now consider spacetime with a boundary and set $M_{D+1} = M_D \times I$ with the left and right boundary conditions $\mathscr{B}^{\mathrm{sym}}$ and $\mathscr{B}^{\mathrm{phys}}$, respectively. As mentioned, the $\mathscr{B}^{\mathrm{sym}}$ describing $\mathcal{S} = (D-1)\text{-}\mathsf{Vec}^{\omega}_{\mathbb{G}^{(D)}}$ will satisfy Dirichlet boundary conditions for $c_{r+1}$ and Neumann boundary conditions for $b_{D-r-1}$. This means that $c_{r+1}|_{\mathscr{B}^{\mathrm{sym}}}$ is a background field while $b_{D-r-1}|_{\mathscr{B}^{\mathrm{sym}}}$ remains dynamical.

Notice that the boundary where $c_{r+1}$ is Neumann while $b_{D-r-1}$ is Dirichlet is generally incompatible with the equations of motion due to $\omega$. This is a manifestation of the 't Hooft anomaly characterized by $[\omega]$ that prevents the anomalous parts of $\mathbb{G}^{(D)}$ from being gauged.

When the symmetry is anomaly-free, the simplest topological defects of the symTFT are

$$T_{r+1}^{(c)}(M_{r+1}) = e^{\frac{2\pi}{N_r} i \int_{M_{r+1}} c_{r+1}}, \tag{C8}$$

$$T_{D-r-1}^{(b)}(M_{D-r-1}) = e^{\frac{2\pi}{N_r} i \int_{M_{D-r-1}} b_{D-r-1}}, \tag{C9}$$

and they satisfy

$$\langle T_{r+1}^{(c)} T_{D-r-1}^{(b)} \rangle = e^{\frac{2\pi i}{N_r} \operatorname{link}(M_{r+1}, M_{D-r-1})}. \tag{C10}$$

$T_{r+1}^{(c)}$ can end on the aforementioned $\mathscr{B}^{\text{sym}}$ boundary, while $T_{D-r-1}^{(b)}$ remains dynamical topological defects on this boundary. Thus, the symmetry category is formed by the topological defects $T_{D-r-1}^{(b)}$ and their condensation defects. On this boundary, $T_{D-r-1}^{(b)}$ transforms the ends of $T_{r+1}^{(c)}$ according to Eq. (C10), and therefore generates the anomaly-free version of the symmetry Eq. (C3).

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
