# Peer review of "Emergent generalized symmetries in ordered phases"

_SciPost Physics_

## Round 3 · Referee Report · Anonymous (Referee 1) · 2024-2-12

Report

The manuscript discuss emergent symmetries in ordered phase with spontaneously symmetry breaking. The emergent symmetries are organized in terms of homotopy groups of the sigma model. Before I can recommend it for publication, here are a few questions to be addressed: - The author does not discuss possible topological action such as theta term or Wess-Zumino term: even when the symmetry breaking pattern is the same, there are different sigma models distinguished by topological actions, and they can have different symmetries.

  • The author discuss symmetry in terms of homotopy groups instead of cohomology. However, homotopy group does not always give the correct symmetry, see e.g. https://arxiv.org/pdf/1707.05448.pdf https://arxiv.org/abs/2210.13780

  • The author discuss whether homotopy defects are invertible. But fusing two homotopy defects can produce nontrivial non-topological defects with topological charge zero (e.g. most elementary excitations have zero topological charges). Can the author clarify how the fusion is defined?

  • There is a discussion using Postnikov system. What is the physical meaning in terms of defects, e.g. does it imply some relations between correlation function? (as the defects are generally not topological, it is hard to imagine there is such universal relation just from homotopy groups)

Requested changes

See the questions in the report

---

## Round 3 · Referee Report · Anonymous (Referee 2) · 2024-3-19

Report

This article presents an interesting perspective on where generalized symmetries can appear in quantum systems. The author describes how ordered systems of conventional or more generally invertible/higher group symmetries could provide a pretty general avenue to find more exotic higher categorical symmetries. While this is an interesting and less explored work in the literature thus far, I think the work could benefit from having more examples formulated as conventional quantum field theories and Hamiltonian lattice models to convey the theoretical ideas.

Before recommending this work, I also have some slightly more specific questions:

  1. Can the author describe the structure of 3Rep(G) symmetries concretely in some G symmetry broken phase. This category has infinitely many simple objects that presumably act identically on the charged operators. How are these simple objects represented within a concrete model?

  2. In examples where the symmetry is 2-group with a non-trivial Postnikov class, how does the Postnikov class appear in the properties of the homotopy defects?

  3. What are the forms of the condensation defects in 2Rep(S3) or 2Rep(D8) concretely realised within an ordered model? How do these defects act the homotopy defects?

  4. Similarly is it possible to write the form of Q8 non-invertible 1-form symmetry generators, say wrapping a non-contractible cycle of space, in a concrete model displaying SO(3)—> Z2 x Z2 SSB? What determines the vacuum expectation value of such an operator?

---

## Editorial Decision

resubmitted